# Brain methylome remodeling selectively regulates neuronal activity genes linking to emotional behaviors in mice exposed to maternal immune activation

Li Ma[1,5], Feng Wang[2,5], Yangping Li[2,5], Jing Wang[3,5], Qing Chang[1], Yuanning Du [1], Jotham Sadan[1], Zhen Zhao [4], Guoping Fan[3] ✉, Bing Yao [2] ✉ & Jian-Fu Chen [1] ✉

How early life experience is translated into storable epigenetic information leading to behavioral changes remains poorly understood. Here we found that Zika virus (ZIKV) induced-maternal immune activation (MIA) imparts offspring with anxiety- and depression-like behavior. By integrating bulk and single-nucleus RNA sequencing (snRNA-seq) with genome-wide 5hmC (5-hydroxymethylcytosine) profiling and 5mC (5-methylcytosine) profiling in prefrontal cortex (PFC) of ZIKV-affected male offspring mice, we revealed an overall loss of 5hmC and an increase of 5mC levels in intragenic regions, associated with transcriptional changes in neuropsychiatric disorder-related genes. In contrast to their rapid initiation and inactivation in normal conditions, immediate-early genes (IEGs) remain a sustained upregulation with enriched expression in excitatory neurons, which is coupled with increased 5hmC and decreased 5mC levels of IEGs in ZIKV-affected male offspring. Thus, MIA induces maladaptive methylome remodeling in brain and selectively regulates neuronal activity gene methylation linking to emotional behavioral abnormalities in offspring.

Maternal immune activation (MIA) refers to as the activation of a pregnant woman's immune system during pregnancy, which can be triggered by various factors, including infections, autoimmune diseases, or exposure to certain environmental factors[1]. Epidemiological studies have revealed a link between MIA and neuropsychiatric disorders that may develop in their children, including autism spectrum disorder (ASD)[2,3], schizophrenia[4], and bipolar disorder[1,5]. Animal models corrate this link and show that MIA can lead to abnormal neuropathology and behaviors in offspring later in life[1]. These MIA animal models are elicited by exposing during gestation to specific infectious agents including influenza virus or immune-activating agents such as the viral mimic polyriboinosinic-polyribocytidylic acid [poly(I:C)] or the bacterial endotoxin lipopolysaccharide (LPS). Converging evidence from these animal models show that maternal immune activation induces adverse neurocognitive behaviors in offspring, which can be transmitted to the next generation[6]. However, how maternal infection as an environmental risk factor shapes the neuropathological and behavioral outcomes in offspring later in life remains to be fully elucidated.

The recent outbreak of Zika virus (ZIKV) in 2015-2016 continues to fuel research interest in maternal infection. Maternal ZIKV infection leads to congenital Zika syndrome (CZS), with phenotypic outcomes

[1]Center for Craniofacial Molecular Biology, University of Southern California, Los Angeles, CA 90033, USA. [2]Department of Human Genetics, Emory University School of Medicine, Atlanta, GA 30322, USA. [3]Department of Human Genetics, David Geffen School of Medicine, University of California Los Angeles, Los Angeles, CA 90095, USA. [4]Zilkha Neurogenetic Institute, Keck School of Medicine, University of Southern California, Los Angeles, CA 90033, USA. [5]These authors contributed equally: Li Ma, Feng Wang, Yangping Li, Jing Wang. ✉e-mail: guopingfan@gmail.com; bing.yao@emory.edu; Jianfu@usc.edu

dependent on infection timing. ZIKV infection during the first or second trimester of pregnancy tends to result in a wide spectrum of structural abnormalities including microcephaly, fetal growth restriction, intracranial calcification, stillbirth, ocular disorders, and craniofacial disproportion, among others[7–9]. In contrast, ZIKV infection during the third trimester of pregnancy typically does not result in microcephaly in offspring. It has been estimated that the majority of live infants born to ZIKV-positive women exhibit no overt clinical manifestations of CZS at birth[10,11]. However, some *in utero* ZIKV-exposed infants with normal head sizes at birth start to exhibit a range of clinical symptoms including seizures/epilepsy later in life[12,13]. An investigation of children aged 12 to 32 months who were born during the ZIKV epidemic in 2015-2016 revealed an increase in delayed neurodevelopment and neurosensory alterations[14]. Epidemiological studies of infants born in 2016-2017 detected general movement abnormalities associated with prenatal ZIKV infection[15]. Overall, the long-term behavioral effects in offspring exposed to maternal ZIKV infection remain poorly understood.

Multiple mechanisms have been shown to contribute to the pathogenesis of behavioral abnormalities associated with maternal infection[1,16]; among them, epigenetic regulation has been highlighted. Epigenetic regulatory mechanisms produce heritable changes in gene expression without alterations to the DNA sequence, and include post-translational histone and DNA modifications. DNA modifications primarily refer to covalent chemical modifications to the 5-carbon position of cytosine, such as the addition of a methyl group to create 5-methylcytosine (5mC) and the further oxidation from 5mC to 5-hydroxymethylcytosine (5hmC), both of which play critical and distinctive epigenetic roles in brain development and diseases[17]. 5mC is generally associated with repressing gene expression when promoter CpG islands are methylated. In contrary, 5hmC, oxidized from 5mC by ten-eleven translocation (TET) proteins[18,19], is primarily associated with gene activation. Emerging evidence suggests that maternal infection can induce stable changes in offspring DNA methylation coupled with alterations in gene expression[20,21]. However, neither 5hmC nor 5mC have been examined in CZS; more broadly, it remains unknown whether 5hmC is altered in offspring exposed to maternal infection, leading to a functional impact on gene expression.

We recently established a mouse model of cognitive behavioral abnormalities associated with MIA by ZIKV infection during pregnancy[22], which recapitulates neurocognitive behavioral symptoms observed in human CZS[14,23]. We performed the neural circuit study of CZS and identified the hyperactive long-range circuit from the ventral hippocampus (vHIP) to the medial prefrontal cortex (mPFC) that drives social memory deficits in ZIKV offspring mice[22]. Here we investigated molecular mechanisms underlying dysconnectivity and behavioral abnormalities in ZIKV offspring mice. By integrating genome-wide 5hmC and 5mC profiling with bulk and snRNA-seq, we found that methylome remodeling leads to sustained, rather than transient, activation of neuronal activity-dependent IEGs, contributing to behavioral abnormalities in ZIKV offspring mice. Overall, our study defines an epigenetic mechanism that illustrates how early maternal inflammation as a transient adverse environmental factor leads to long-term neural connectivity and behavioral abnormalities later in offspring life.

## Results

### Anxiety and depressive behaviors in offspring mice exposed to maternal immune activation (MIA) by ZIKV infection

To model ZIKV transmission from mosquitoes to humans, we administered 200 μl of $1.7 \times 10^5$ TCID50/ml (low dose, referred to as ZIKV[low]) or $1.7 \times 10^6$ TCID50/ml (high dose, referred to as ZIKV[high] or ZIKV) virus (Mexican isolate MEX1-44) by intravenous injection to pregnant mice at embryonic day 12.5 (E12.5). Vehicle (PBS) injection was used as a mock control. Maternal immune response was efficiently activated by both the low and high dose of ZIKV, as evidenced by significantly increased serum concentration of IL-6 and TNF-a at 3 hours after virus injection (Fig. 1a-c), which also resulted in a strong increase of serum IL-17a at E14.5 (Fig. 1a, d). Notably, ZIKV[high] infection could induced more robust immune response than ZIKV[low] and was comparable to poly(I:C), a synthetic analogue of double-stranded RNA, which was widely used to induce maternal immune response[1,24]. ZIKV-infected dams exhibited decreased activity in the open field and burrowing behavior at 4 and 8 hours post-infection (Supplementary Fig. 1a, b). The body weight did not increase until 24 hours post-infection (Supplementary Fig. 1c), which were typical behavioral and physiological signs of sickness. The poly(I:C) and ZIKV[high] infection resulted in ~20% abortion rate (Supplementary Fig. 2a), while had no obvious impact on pregnancy outcomes of dams, which successfully delivered pups (Supplementary Fig. 2b-e). We also did not detect obvious morphological defects in offspring mice after maternal ZIKV exposure. As demonstrated in our published studies[22], we cannot detect ZIKV in the placenta or fetal brains after this infection protocol, but the ZIKV infection during pregnancy triggers MIA and induces an inflammatory response in the fetal brains.

A battery of behavioral tests was performed on the offspring at postnatal 8-9 weeks. In an open field test (Fig. 1e), ZIKV-affected offspring mice spent less time in the center and traveled shorter distances compared to mock controls (Figs. 1e, g–h). These phenotypes occurred after ZIKV[high] but not ZIKV[low] infection, suggesting ZIKV dose-dependent anxiety in offspring mice exposed to maternal ZIKV infection. To confirm the presence of anxious behaviors, we next performed an elevated plus maze test (Fig. 1f). ZIKV[high] offspring mice spent less time in open arms and made significantly fewer entries into open arms compared to controls (Figs. 1i-j). ZIKV[low] offspring mice also made fewer entries into open arms but spent normal amounts of time in open arms (Figs. 1i-j). These results confirmed the presence of anxiety in a ZIKV dose-dependent manner in offspring mice, although ZIKV[low] offspring mice exhibited relatively mild phenotypes. Therefore, we focused on ZIKV[high] offspring mice for our further studies (ZIKV[high] is referred to as ZIKV hereafter). To examine whether offspring mice exhibited depressive behaviors, we performed a forced swimming test and found a significant increase in immobility time of ZIKV offspring mice compared to controls (Fig. 1k), suggesting a depression-like behavioral phenotype. To determine if these behavioral abnormalities were due to reduced motor strength, we performed a rotarod assay and did not find a significant difference in average latency to fall between ZIKV and control groups (data not shown), suggesting normal motor strength in ZIKV offspring. To measure motor coordination and motor learning, we next performed an accelerating rotarod test. Control mice exhibited an increased trend of latency to fall in the rotarod test over a 4-day test period, indicating an active learning process during this period of time. In contrast, ZIKV offspring mice displayed a slight but significant delay in motor learning (Fig. 1l). We did not find a significant difference in animal behaviors among different sexes (Fig. 1m, Supplementary Fig. 3). Our previous studies showed that ZIKV offspring mice exhibited autism-like cognitive deficits[22]; current work identified emotional deficits including anxiety and depressive-like behaviors in offspring mice exposed to maternal ZIKV infection.

### IEG upregulation in brains of adult ZIKV offspring mice

To investigate mechanisms underlying emotional behavioral abnormalities in ZIKV offspring mice, we performed bulk RNA-sequencing analysis of 2-month-old mouse prefrontal cortex (PFC) tissues. We focused on PFC because this region plays essential roles in controlling cognitive and emotional behaviors[25], which are disrupted in ZIKV offspring mice (Fig. 1, Supplementary Fig. 3)[22]. We also detected altered excitation and inhibition (E/I) balance coupled with cortical hyperexcitability in PFC of ZIKV offspring mice[22]. E/I imbalance is

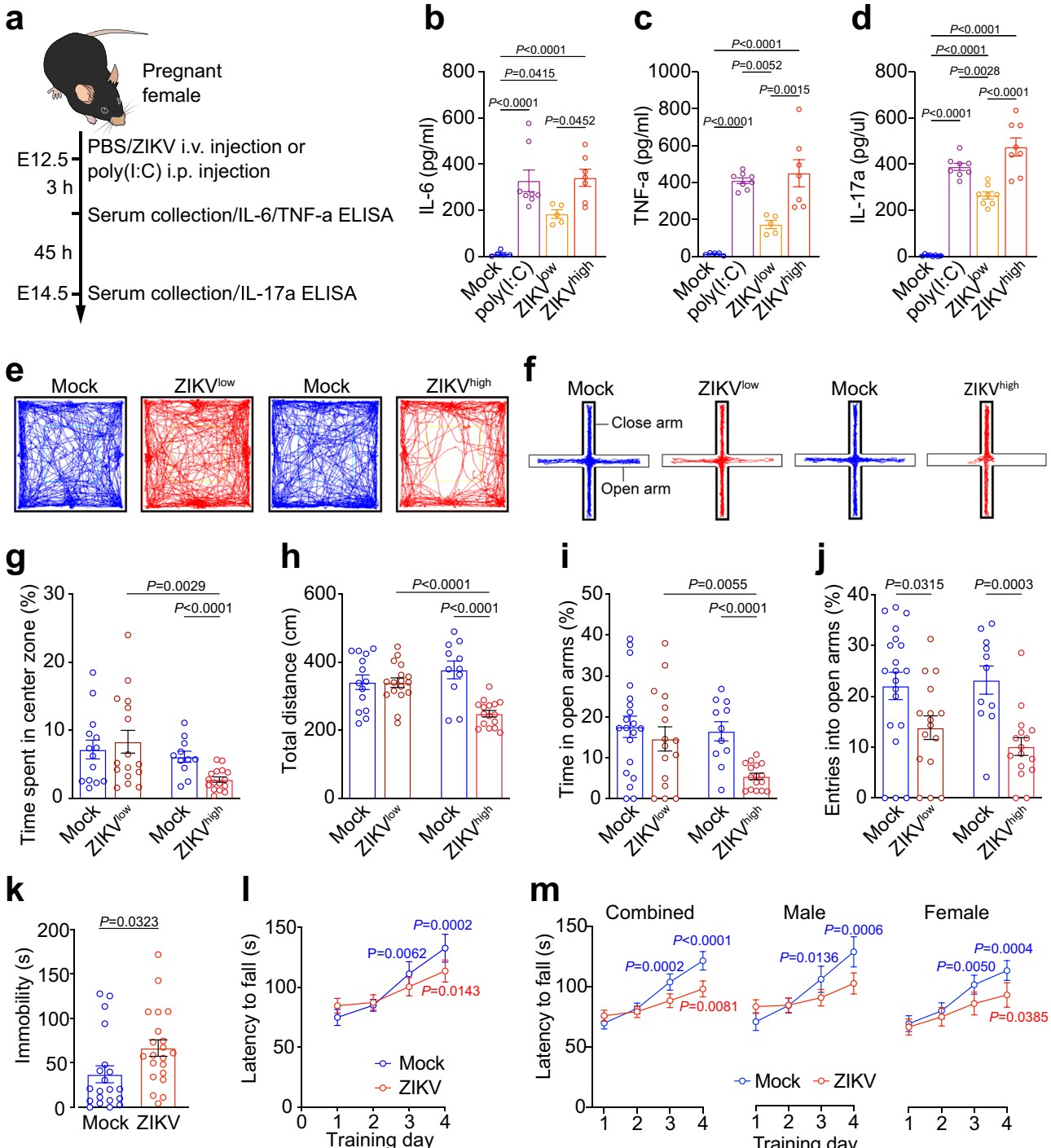

**Fig. 1 | Anxiety- and depression-like behaviors in offspring mice exposed to ZIKV-induced MIA. a** The schematic showing the experimental procedure for ELISA. **b–d** ELISA measurement of serum concentrations of maternal IL-6 (**b**), TNF-a (**c**) and IL-17a (**d**). **b** and **c**, Mock $n = 5$; poly(I:C) $n = 8$; ZIKV[low] $n = 5$; ZIKV[high] $n = 7$ mice. **d**, Mock $n = 8$; poly(I:C) $n = 8$; ZIKV[low] $n = 8$; ZIKV[high] $n = 8$ mice. **e**, **f** Representative animal tracks in open field (**e**) and elevated plus maze tests (**f**). **g** Percentage of time spent in the center zone during open field test. Mock $n = 14$, ZIKV[low] $n = 16$; Mock $n = 11$, ZIKV[high] $n = 16$ mice. **h** Total distance traveled in the open field test. Mock $n = 14$, ZIKV[low] $n = 17$; Mock $n = 11$, ZIKV[high] $n = 16$ mice. **i** Time spent in the open arms in the elevated plus maze. Mock n = 20, ZIKV[low] $n = 16$; Mock $n = 11$, ZIKV[high] $n = 16$ mice. **j** Number of entries into open arms during elevated plus maze

test. Mock $n = 20$, ZIKV[low] $n = 16$; Mock $n = 11$, ZIKV[high] $n = 16$ mice. **k** Immobility time in the forced swimming test. Mock $n = 20$, ZIKV[high] $n = 21$ mice. **l** Rotarod performance scored as time (seconds) on the rotarod. Mock $n = 13$, ZIKV $n = 14$ mice. **m** Rotarod performance scored as time (seconds) on the rotarod. Combined-Mock $n = 23$; Combined-ZIKV $n = 26$; Male-Mock $n = 12$; Male-ZIKV $n = 14$; Female-Mock $n = 11$; Female-ZIKV $n = 12$ mice. Representative results were obtained from at least three independent experiments with similar results. All data are presented as mean values ± SEM. $P$ values were calculated by one-way ANOVA with Tukey post hoc tests (**b–d**) and two-tailed unpaired t test (**g–m**). Source data are provided as a Source Data file.

sufficient to cause cognitive and emotional behavioral alteration in ASD patients[26]. Pairwise differential expression analysis of the PFC transcriptome of control and ZIKV offspring mice identified differentially expressed genes (DEGs) in the ZIKV group, with mean of normalized counts > 150 and threshold of $\log_2$ fold change < -0.15 (downregulated) or > 0.15 (upregulated) (Fig. 2a). We observed 469 down-regulated genes and 632 up-regulated genes in the ZIKV group (Fig. 2a). Gene Ontology (GO) analysis showed that upregulated DEGs in ZIKV PFC were significantly enriched in functional clusters including nervous system development, developmental process, and system development (Fig. 2b). Thus, ZIKV-triggered early maternal inflammation has a long-term consequence on transcriptomic dysregulation in adult offspring mouse PFC.

Strikingly, we noticed that many top-upregulated genes in ZIKV offspring brains belong to the class of neuronal-activity-dependent IEGs, such as *Egr1-4*, *Arc*, *Fos*, *Nr4a1*, and *Junb*, among others (Fig. 2a). IEGs are quickly activated by neural activity and then are rapidly inactivated under normal conditions[27]. Past research has established that IEGs integrate neuronal activities with adaptive behaviors under physiological and pathological (mostly genetic mutation) conditions[28,29]. However, the regulation and function of IEGs in maternal inflammation remain unexplored. We reasoned that sustained IEG upregulation could be responsible for cortical hyperexcitability leading to cognitive and emotional behavioral abnormalities. To test this hypothesis, we performed RT-PCR and validated the IEG upregulation in ZIKV offspring PFC tissues, including *Egr1-4*, *Arc*, *Fos*, *Npas4*, *Nr4a1*, *Junb*, *Dusp1*, *Fosb*, and *Fosl2* (Fig. 2c). Immunohistochemical (IHC) staining of PFC tissues further confirmed the increase in the expression of IEGs *Fos*, *Npas4*, and *Egr1* in ZIKV offspring mice with or without social exposure to the stranger mouse (Figs. 2d-g). Together, these experiments identified increased IEGs as the top upregulated genes in PFC of mice exposed to maternal inflammation due to ZIKV.

## The snRNA-seq analysis of ZIKV offspring mouse brains

To investigate cell type-specificity of IEG upregulation, we next performed snRNA-seq analysis of PFC. We selected the single-nucleus approach as opposed to the single-cell RNA-seq approach because it is technically challenging to effectively isolate viable single cells of rare cell populations from adult mouse brains. We collected PFC tissue (AP + 1.3 mm - +2.2 mm, 300 μm thickness) from control and ZIKV groups (*n* = 4 per group) to control for variability in the dissection and other individual variation (Fig. 3a). We captured about 16,135 single nuclei and sequenced a median of 1,096 genes per nucleus. Considering that nuclear RNAs were profiled, 42% of unique molecular identifiers (UMIs) were mapped to exons and 31% to introns; therefore, the gene expression profiles of nuclei likely reflected nascent transcripts and the cellular transcriptome. After unbiased clustering of nuclear profiles from ZIKV and control samples, cell-type identity was defined based on the top DEGs and expression of known cell-type marker genes (Fig. 3b). We identified 15 primary cell types (Fig. 3c), including five types of excitatory neurons, four types of inhibitory neurons, astrocytes, oligodendrocytes, microglia, oligodendrocyte precursor cells (OPCs), and endothelial cells. These cell cluster markers and their constituent cell types were well-separated (Fig. 3c), suggesting that our snRNA-seq data were of high integrity.

There were no significant changes in the numbers of individual cell types between control and ZIKV offspring brains (Fig. 3d); in both groups, excitatory neurons accounted for the most abundant cell types (Fig. 3d). These results suggest that there was no significant reduction of specific cell types, which is consistent with the normal brain morphology of ZIKV offspring mice. Next, we examined DEGs in each cell type between the control and ZIKV groups. In total, we identified ≥145 DEGs (FDR < 0.01) in one or more cell types. By downsampling the data for burden analysis, we found that In4

(NPY) inhibitory neurons had the largest number of DEGs, followed by Ex3 (CALM2) and Ex5 (TSHZ2) excitatory neurons, with the lowest number of DEGs in endothelial cells (Fig. 3e). Hierarchical clustering based on log-transformed relative (fold) changes (ZIKV vs. Mock) of DEGs in each cell type revealed that those dysregulated genes are associated with different brain development and function, including neurogenesis, learning or memory, ion transport, and neurotransmitter transport, among others (Fig. 3f). Importantly, snRNA-seq confirmed the IEG upregulation in ZIKV offspring PFC and further revealed that this upregulation was enriched in the excitatory neurons (Fig. 3g). In summary, snRNA-seq revealed IEG upregulation in specific neuronal types in PFC of adult mice exposed to MIA by ZIKV.

## Genome-wide 5hmC remodeling in ZIKV offspring mouse brains

Epigenetic regulation mediates environmental influences on genome architecture and gene expression[8,24]. In contrast to DNA methylation, 5hmC-mediated DNA demethylation has not yet been investigated in the context of maternal inflammation. To understand the epigenetic mechanism underlying IEG upregulation and behavioral abnormalities in ZIKV offspring mice, we sought to systematically identify 5hmC dynamics from control and ZIKV PFC samples. The 5hmC reads were identified by a previously established chemical labeling and affinity purification method coupled with high-throughput sequencing technology[30], and were normalized in 10 kb binned mouse genome (mm9). Using 5hmC normalized read density in genome-wide unbiased 500 bp bins, linear regression analysis suggested a global loss of 5hmC in the PFC of ZIKV offspring mice (Fig. 4a, *P* < 2.2e-16). Given mounting evidence suggesting that 5hmC on the gene body is positively correlated with gene expression[31], we first assessed 5hmC levels on the gene body and found that intragenic 5hmC was significantly reduced in ZIKV groups compared to controls (Fig. 4b upper panel, *P* = 6.30e-07). Furthermore, 5hmC levels were also reduced in previously defined cortex-specific enhancers[31] (Fig. 4b lower panel, *P* = 0.007). Interestingly, while global 5hmC decreased upon maternal ZIKV infection, transposon elements (TEs) including SINEs and LINEs exhibited increased 5hmC, and ~93.42% of repetitive elements showed 5hmC accumulation in ZIKV offspring mice (Fig. 4c). To correlate 5hmC levels with gene expression, we analyzed bulk RNA-seq data and revealed dynamically up- or down-regulated expression of each type of repetitive element in ZIKV offspring (Fig. 4d). Given that dysregulation of repetitive elements has been strongly linked to mental illnesses such as depression and anxiety disorders[32], 5hmC-mediated dysregulation of repetitive elements could contribute to behavioral abnormalities in mice exposed to MIA by ZIKV.

Using a published rigorous differential analysis algorithm[33], we identified two groups of significantly differentially hydroxymethylated regions (DhMRs), with 1300 gain-of-5hmC and 1985 loss-of-5hmC regions in response to ZIKV maternal inflammation (FDR < 0.05, Fig. 4e). Annotation of those regions revealed that most gain-of-5hmC changes were intergenic or intronic, whereas a substantial number of loss-of-5hmC regions were located in exons (Fig. 4e, right panels). These observations are supported by detailed Integrative Genomic Viewer (IGV) views of representative DhMRs identified from our 5hmC profiling. For example, the indicated DhMR in *Entpd4*, a gene that has been linked to sleep-related disorders and psychiatric diseases[34], showed significant loss of 5hmC on its exons (Fig. 4f). On the other hand, intronic gain of 5hmC was found in *Dgki* (Fig. 4f), a member of the type IV diacylglycerol kinase subfamily that may play a role in presynaptic diacylglycerol/DAG signaling and control neurotransmitter release during metabotropic glutamate receptor-dependent long-term depression[35]. Both genes showed a positive correlation between intragenic 5hmC and gene expression. Together, these results showed that the genomic distribution of 5hmC is remodeled in the PFC of ZIKV offspring mouse brains.

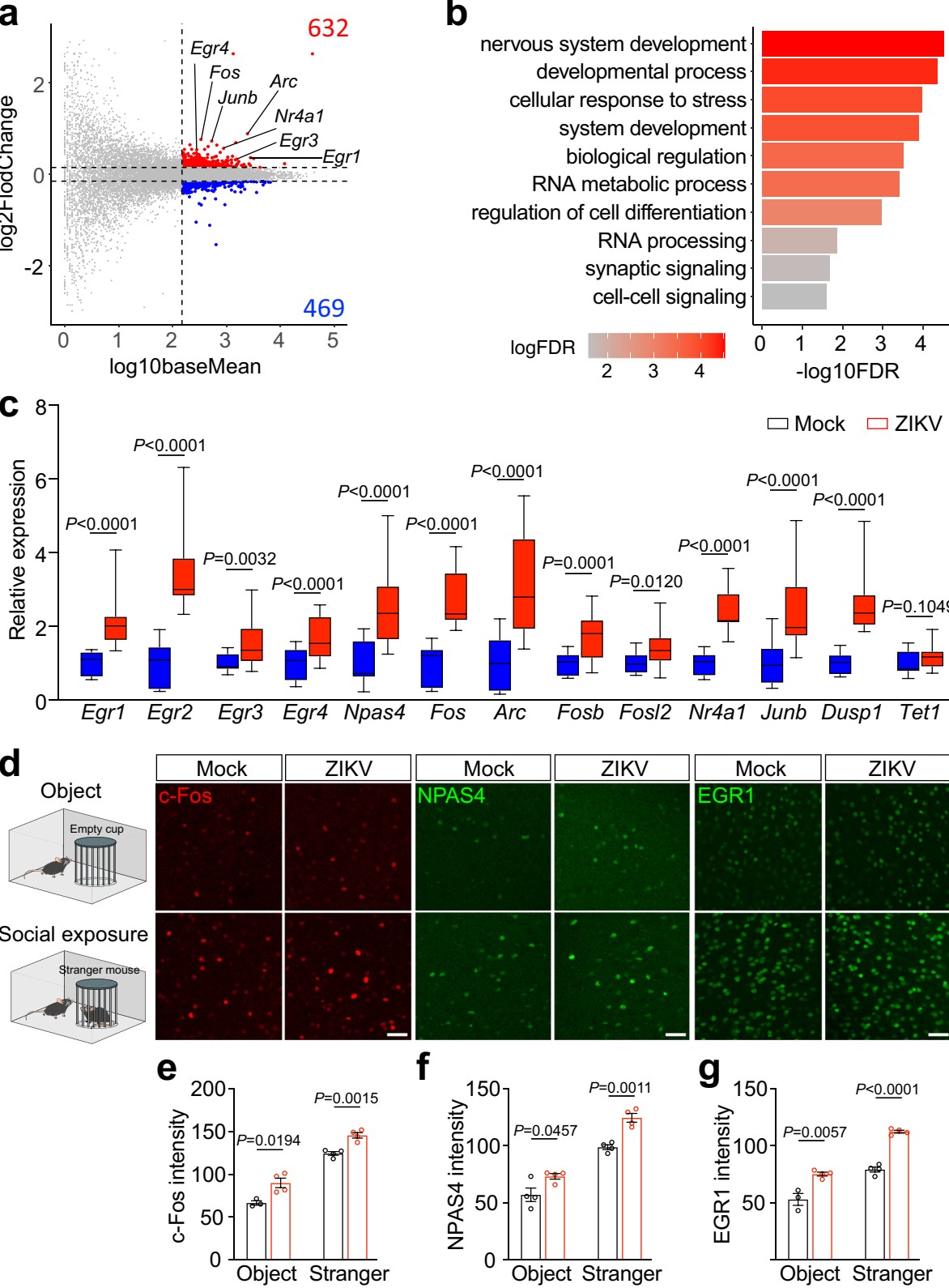

**Fig. 2 | Immediate-early gene (IEG) upregulation in ZIKV offspring mice. a** Log2-fold change and average expression level of all genes quantified by bulk RNA-seq reads in the PFC of ZIKV offspring mice. **b** Gene ontology (GO) analysis for significantly upregulated genes in ZIKV-affected offspring mice. **c** RT-PCR analysis of IEG expression in the PFC tissues of control and ZIKV offspring mice. Box−whisker plot (displaying the Max/Min at the whiskers, the 75/25 percentiles at the boxes, and the median in the center line). $n = 18$ biological replicates per group. **d** IHC staining of c-Fos, NPAS4, and EGR1 in PFC tissues of control and ZIKV offspring mice with or without a social exposure to another mouse. Three independent experiments were repeated with similar results. Scale bar, 50 μm. **e**−**g** Quantification of the intensity of c-Fos, NPAS4, and EGR1. For **e**−**g**, $n = 3$-4 biological replicates per group and each data point represents one biological replicate. Data are means ± SEM. $P$ values were calculated by two-tailed unpaired t test (**c**, **e**−**g**). Source data are provided as a Source Data file.

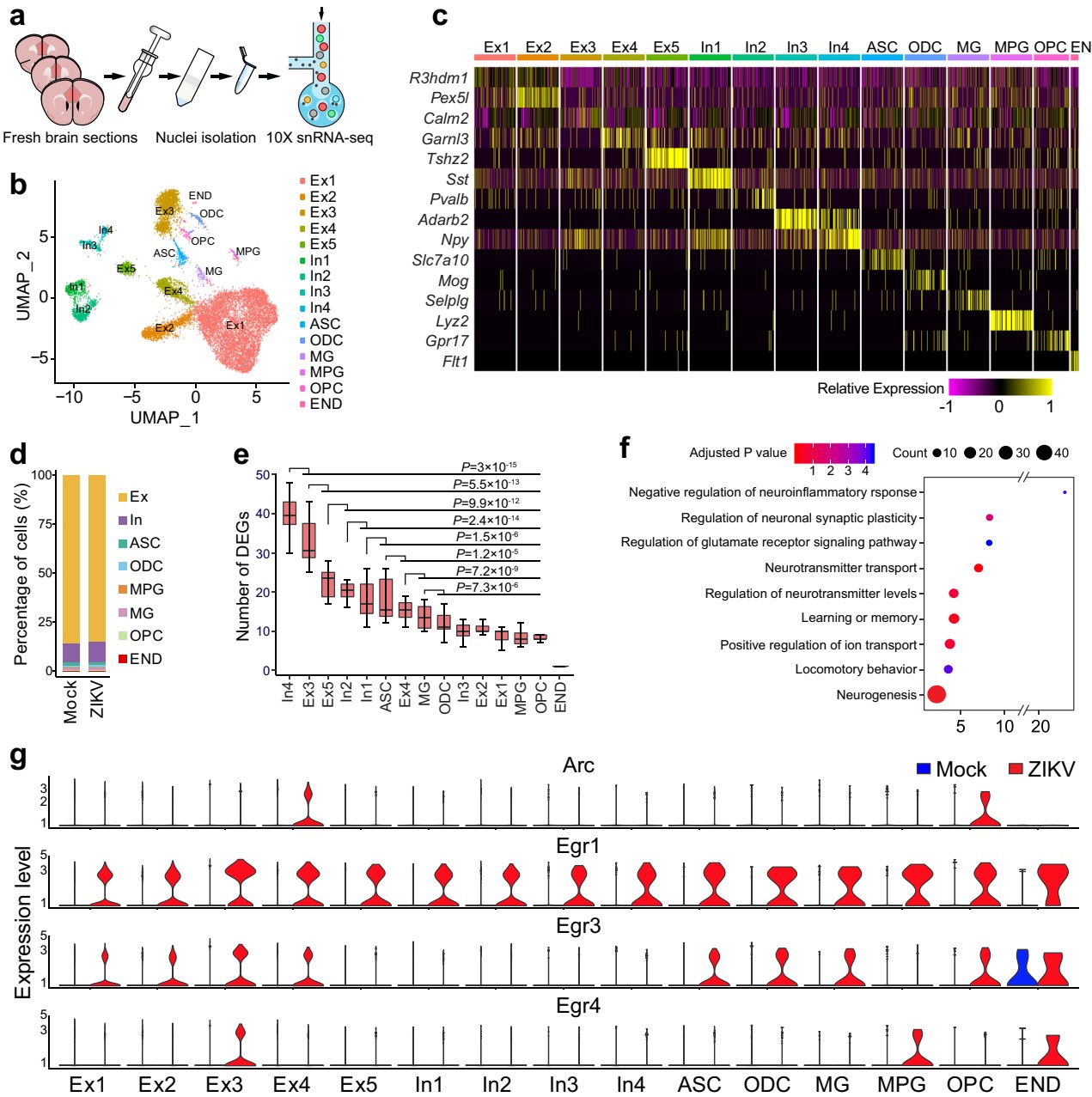

**Fig. 3 | Single nucleus RNA sequencing (snRNA-seq) analysis of PFC in ZIKV offspring mice. a** The experimental scheme for snRNA-seq. **b** Expression of markers for each cell subtype in the mouse PFC. **c** 15 clusters across 16,135 cells from all cell types in cortex on a UMAP visualization. **d** Proportion of each cell cluster in ZIKV group vs. mock control. **e** Burden analysis of numbers of differentiated expressed genes (DEGs) in individual cell clusters. Box–whisker plot (displaying the Max/Min at the whiskers, the 75/25 percentiles at the boxes, and the median in the center line). $N = 10$ biological replicates per cell type. $P$ values were calculated by

two-tailed unpaired $t$-test. **f** GO analysis of DEGs in PFC of ZIKV offspring vs. mock control. DEGs refer to genes with consistent fold changes between snRNA-seq and bulk RNA-seq. The $P$ value of the overrepresentation of a gene set was calculated using a Fisher's exact test (FET) in each of these gene lists. **g** Violin plots showing differently expressed IEGs, including *Egr1, 3, 4* and *Arc* in each cell subtype of the mouse PFC. Note the enriched IEG upregulation in excitatory neurons. Source data are provided as a Source Data file.

## 5hmC alterations correlate with expression changes of key genes in neuropsychiatric disorders

To investigate the functional impact of 5hmC changes on gene expression, we analyzed genome-wide 5hmC profiling and bulk RNA-seq data together. Indeed, the intragenic 5hmC changes were correlated with transcriptome changes in the PFC of ZIKV offspring mice. A total of 692 downregulated genes with at least one significant loss-of-5hmC region and 1285 upregulated genes with significant gain-of-5hmC regions in their gene bodies were identified (Fig. 5a, blue and orange dots, respectively). Gene Ontology (GO) analysis revealed that

many genes with concomitant downregulation of 5hmC and gene expression were related to neuronal development and functions (Fig. 5b). In contrast, genes that exhibited simultaneous upregulation of 5hmC and mRNA expression were enriched in the metabolic process and protein modification categories (Fig. 5c). To further validate the 5hmC and gene expression from genomic data, we carried out an orthogonal analysis by performing the 5hmC-capture experiments followed by qPCR analysis using primers targeting several DhMRs. Real-time qPCR data showed consistent 5hmC changes in these loci as that in 5hmC-seq data (Fig. 5d). To validate the bulk RNA-seq data, we

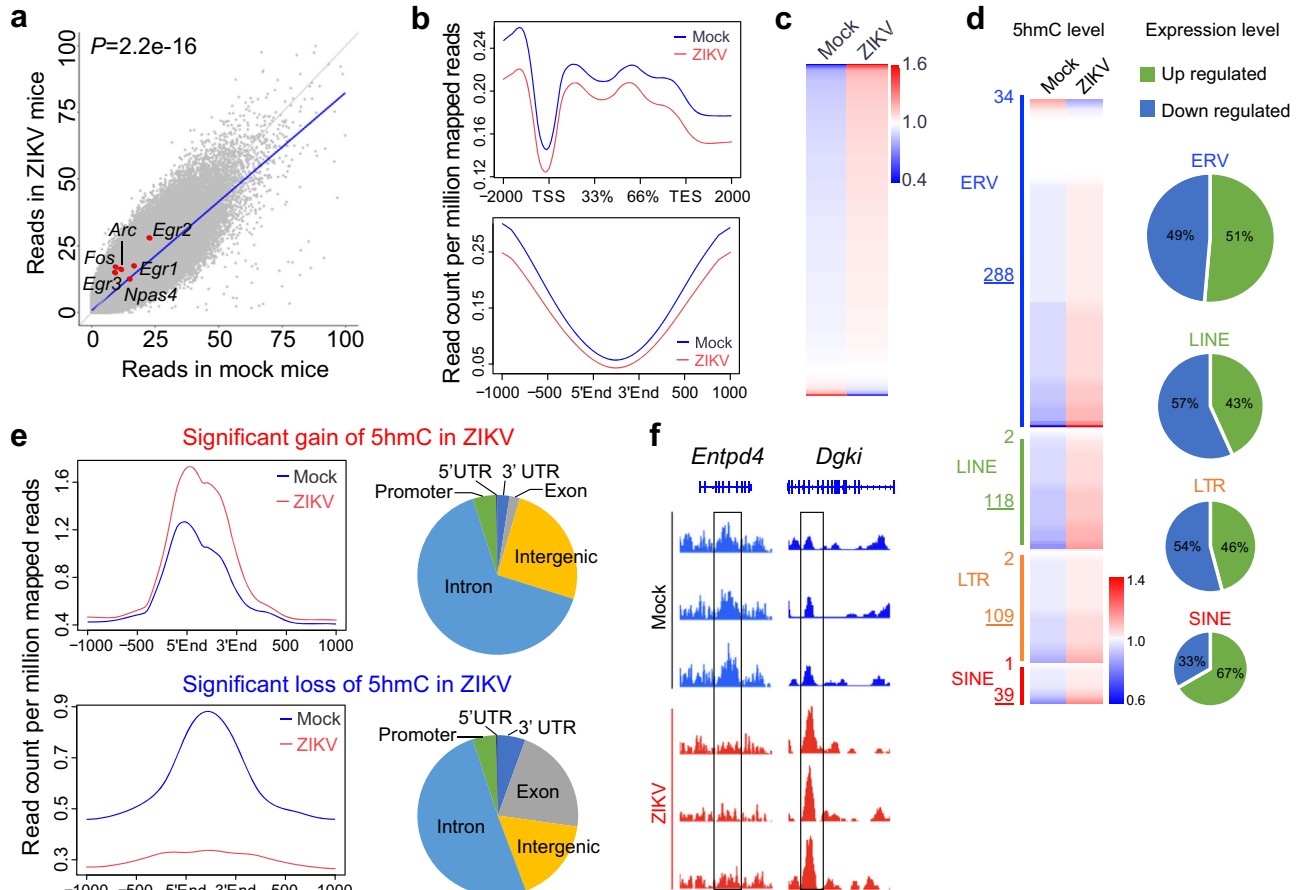

**Fig. 4 | Genome-wide 5hmC changes in PFC of ZIKV offspring mice. a** Normalized read counts in ZIKV-exposed offspring mice and controls of each 500 bp bin. The fitted linear regression model shown as a blue line indicated a genome-wide loss of 5hmC in ZIKV-exposed mice ($R = 0.64$, $P < 2.2e-16$). $P$ values were calculated by Pearson correlation. **b** Average 5hmC level in gene body (upper panel) for all mouse genes showed loss of 5hmC in gene bodies ($P = 6.30e-07$, Fold change=0.88). Average 5hmC in mouse cortex enhancer (lower panel) showed loss of 5hmC in cortex enhancer region ($P = 0.007$, Fold change=0.81). $P$ values were calculated by two-tailed unpaired t test. **c** Heatmap for 5hmC level of each repetitive element showed gain of 5hmC in most repetitive elements. **d** Gain-of-5hmC is accumulated in most repetitive elements in PFC of offspring mice exposed to maternal ZIKV infection. Pie chart shows expression changes of repetitive elements with gain-of-5hmC regions. **e** Average 5hmC level and genomic distribution feature for gain-of-5hmC regions in ZIKV-exposed mice (upper panels); Average 5hmC level and genomic distribution feature for loss-of-5hmC regions in ZIKV-exposed mice (lower panels). **f** IGV view of 5hmC changes in *Entpd4* and *Dgki* genes.

selected three upregulated genes (Fig. 5e) with reported biological functions in neuropsychiatric disorders, namely *Slc7a11*, *Mtr*, and *Sox11*. The qPCR analysis confirmed their significant upregulation in the PFC of ZIKV offspring mouse brains (Fig. 5f).

5hmC has been reported to be positively correlated with gene expression[36,37]. We hypothesized that IEG upregulation in ZIKV offspring mouse PFC could be due to altered 5hmC-mediated epigenetic modifications. To test this hypothesis, we analyzed 5hmC profiling data and identified those regions with putative 5hmC level upregulation in individual IEGs in the ZIKV group (Fig. 5g). We next performed 5hmC-capture followed by qPCR analysis using primers targeting these DhMRs of IEGs and found that there is a significant increase of 5hmC in IEGs, including *Egr1*, *Egr3*, *Fos*, and *Arc* (Fig. 5h). To further investigate the cell-type specificity of 5hmC changes in IEGs, we isolated the neurons and non-neuron cells by density gradient centrifugation[38] and the concentrated neurons and non-neuronal cells were confirmed by immunofluorescent staining and FACS analysis using anti-NeuN antibodies (Supplementary Fig. 4a, b). Then the genomic DNA was isolated for MeDIP-seq and 5hmC-seal-seq. IEGs *Egr2*, *Egr3*, *Arc*, and *Fosl2* showed 5hmC accumulation in non-neuronal cells, while *Fosb*, *Npas4*, and *Nr4a1* showed 5hmC accumulation in neuronal cells. *Dusp1*, *Erg4*, and *Junb* showed 5hmC accumulation in both neuronal and non-neuronal cells. (Supplementary Fig. 4c). The IEG upregulation

correlates with increased 5hmC in neuronal or non-neuronal cells (Supplementary Fig. 4d). Together, these results suggest that neuronal and non-neuronal cells undergo cell type-dependent 5hmC changes in IEGs that correlate with their sustained upregulation in offspring cerebral cortex upon ZIKV-induced MIA.

To investigate the functional relevance of 5hmC dynamics in human neuropsychiatric disorders, we attempted to link ZIKV-induced 5hmC modifications with mental illnesses across species. We obtained lists of human genes associated with anxiety, ASD, depression, and schizophrenia, and correlated their mouse orthologs with genes carrying 5hmC changes upon maternal ZIKV infection. Interestingly, the list of genes bearing altered 5hmC in response to maternal ZIKV infection showed significant overlap with all four groups of neuropsychiatric disorder genes (Fig. 5i). In contrast, no significant overlap was found with aortic lesion-associated genes or with obesity-associated genes (Supplementary Fig. 5). Together, these results suggest that 5hmC remodeling might contribute to IEG upregulation as well as dysregulation of other neuropsychiatric disorder genes in ZIKV offspring brains.

### Genome-wide DNA methylation changes in ZIKV offspring mouse brains

5mC DNA methylation changes have been reported in the poly(I:C) MIA mouse model[21]. 5hmC serves as an intermediate step in the

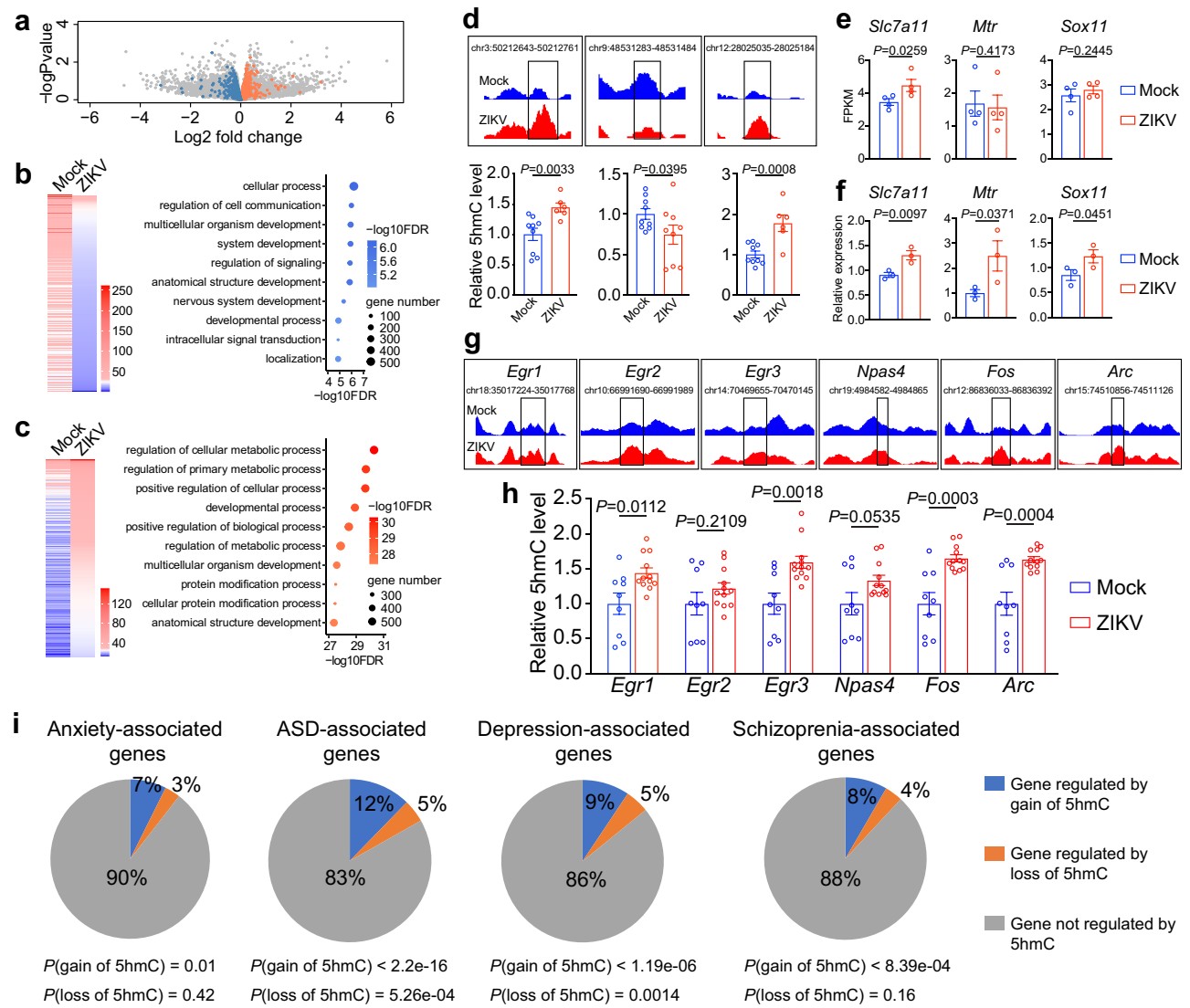

**Fig. 5 | Correlation of 5hmC alterations with expression changes of key genes in neuropsychiatric disorders. a** Gene expression changes in the PFC of offspring mice exposed to maternal ZIKV infection. Downregulated genes with at least one significant loss-of-5hmC are highlighted in blue; upregulated genes with significant gain-of-5hmC are highlighted in orange. RNA-seq log2FoldChange and *P* value were plotted, *n* = 4 biological replicates. DEseq2 were used for 5hmC differential analysis, multiple test adjustment was performed by p.adjust function, method fdr in R, significant 5hmC change was defined by FDR (False Discovery Rate) < 0.05, *n* = 3 biological replicates. **b** Heat map of genes with downregulation of 5hmC and mRNA expression from bulk RNA-seq (left panel); GO analysis of downregulated genes with at least one loss-of-5hmC region (right panel). **c** Heat map of genes with upregulation of 5hmC and mRNA expression from bulk RNA-seq (left panel); GO analysis of upregulated genes with at least one gain-of-5hmC region (right panel). **d** IGV view (upper panel) and qPCR validation (lower panel) of 5hmC genome

profiling data focusing on randomly selected DhMR regions. Chr3:50212643-50212761, mock *n* = 9, ZIKV *n* = 6; chr9:48531283-48531484, mock *n* = 9, ZIKV *n* = 9; chr12:28025035-28025184, mock *n* = 9, ZIKV *n* = 6 biological replicates. *P* values were calculated by one-tailed unpaired *t*-test. **e**, **f** FPKM (Fragments Per Kilobase of transcript per Million mapped reads) from bulk RNA-seq (**e**) followed by qPCR validation of gene expression (**f**). **e** *n* = 4, **f** *n* = 3 biological replicates per group. *P* values were calculated by one-tailed unpaired *t*-test. **g**, **h** IGV view (**g**) and experimental validation of 5hmC levels associated with IEGs (**h**). Mock *n* = 9, ZIKV *n* = 12 biological replicates per gene. *P* values were calculated by two-tailed unpaired *t*-test. **i** Significant overlap between dynamic 5hmC-marked genes in ZIKV offspring mice and genes associated with elevated risk of anxiety, autism spectrum disorder (ASD), depression, and schizophrenia (SCZ). *P*-values are indicated in the figure and were calculated using binomial tests. All data are presented as mean values ± SEM. Source data are provided as a Source Data file.

process of DNA demethylation from 5mC oxidization[18,19], and its genome-wide re-distribution suggested there may be 5mC remodeling in ZIKV offspring mice. Therefore, we profiled genome-wide 5mC DNA methylation and identified 515 differentially methylated regions (DMRs) in the PFC of ZIKV offspring mice compared to controls. There was an overall increase in 5mC DNA methylation in ZIKV offspring mice, reflected by 346 hypermethylation and 169 hypomethylation regions (Fig. 6a, Supplementary Fig. 6a). Unlike Homer annotation for DhMRs (Fig. 4e), hypomethylated regions were mainly annotated to intron and intergenic region while hypermethylated regions were more annotated to promoter (Supplementary Fig. 6b). We used

Principal Component Analysis (PCA) of High-throughput Chromosome Conformation Capture (Hi-C) data to determine whether the chromatin in these regions was in an active or inactive state. We found that these DMRs in ZIKV offspring mouse brains exhibited no significant distribution bias in A/B compartments, suggesting that the DMRs have no preference between active and inactive chromatin (Supplementary Fig. 6c). To understand the genomic context of the DMRs, we investigated their chromatin state by intersecting the DMRs with existing ChromHMM annotations for brain tissues[39]. These DMRs were significantly enriched in the genomic regions characterized as "poised promoter," "strong promoter," and "strong enhancer"

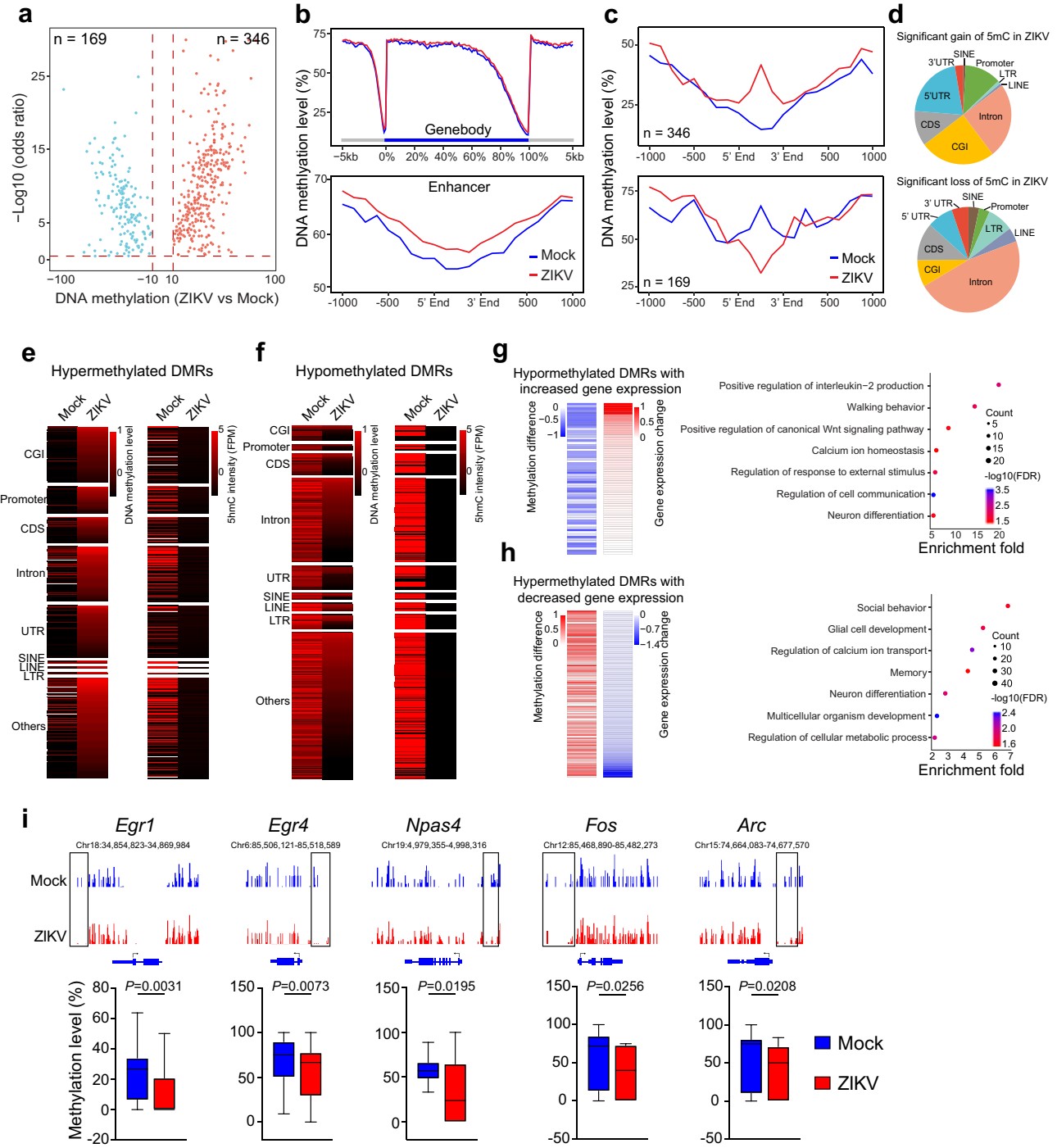

**Fig. 6 | 5mC DNA methylation changes in PFC of ZIKV offspring mice.**
**a** Identification of the DMRs in ZIKV-exposed offspring mice compared to mock controls. **b** Average 5mC DNA methylation levels across the gene body (top) and cortex enhancer (bottom) of all mouse genes. **c**, **d** Average 5mC DNA methylation level and genomic distribution features of gain- and loss-of 5mC regions in ZIKV offspring mice. **e**, **f** Pie charts show that the genome-wide distribution of the hypermethylated DMRs (**e**) or hypomethylated DMRs (**f**) correlates with loss of 5hmC. **g** GO analysis of the up-regulated genes with corresponding hypomethylated DMRs in ZIKV offspring mice. **h** GO analysis of the downregulated genes with

corresponding hypermethylated DMRs in ZIKV offspring mice. **i** IGV view (upper panel) and quantification (lower panel) of 5mC levels associated with IEGs. Box–whisker plot (displaying the Max/Min at the whiskers, the 75/25 percentiles at the boxes, and the median in the center line). *Egr1*, $n = 26$ per group; *Egr4*, $n = 25$ per group; *Npas4*, $n = 14$ per group; *Fos*, $n = 16$ per group; *Arc*, $n = 19$ biological replicates per group. *P* values were calculated by two-tailed unpaired *t*-test. Data are presented as mean values ± SEM. Source data are provided as a Source Data file.

(Supplementary Fig. 6d). Despite the similar levels of average 5mC in their gene bodies, 5mC DNA methylation in enhancers was higher in ZIKV offspring mice than in controls (Figs. 6b-c). We next analyzed genomic features of these gain- and loss-of-5mC DMRs in ZIKV off-spring mice. While hypermethylated DMRs were preferentially located

in the CpG islands (CGIs), intron regions contained significantly enri-ched gain- and loss-of-5mC, representing the highest enrichment associated with maternal ZIKV infection (Fig. 6d). In the non-intronic regions, there were significant differences between gain- and loss-of-5mC genomic distributions. For example, gain of 5mC was

predominantly localized to CGI, CDS, 5′UTR, and promoter regions (Fig. 6d). We next cross-examined 5mC and 5hmC genome profiling data. Interestingly, these hypermethylated DMRs in ZIKV offspring mice displayed fewer 5hmC modifications (Fig. 6e), with enrichment in certain genomic regions such as CGI, promoter, and UTR; these hypomethylated DMRs in ZIKV offspring also had less 5hmC, with enrichment in introns (Fig. 6f).

We analyzed the upregulated genes with hypomethylated DMRs, and GO analysis revealed that these genes are associated with multiple pathways involved in neural development including neuron differentiation, Wnt signaling, and cell communication (Fig. 6g). Similarly, the down-regulated genes with hypermethylated DMRs are linked with social behaviors and neuronal and glial development (Fig. 6h). Our 5mC analysis of IEGs found that there was a decrease of 5mC levels in IEGs including *Egr1, Egr4, Npas4, Fos* and *Arc* (Fig. 6i). We further investigated 5mC changes in IEGs in neuron and non-neuron cells. IEGs *Egr3, Dusp1, Fosb*, and *Egr2* showed 5mC depletion only in non-neuronal cells, while *Nr4a1, Egr1, Fosl2, Junb, Arc*, and *Npas4* showed 5mC depletion in both neuronal and non-neuronal cells (Supplementary Fig. 7a). The IEG upregulation correlates with decreased 5mC in neuron or non-neuron cells (Supplementary Fig. 7b). In both neuron and non-neuron cells, 5hmC positively and 5mC inversely correlate with IEG upregulation, which do not occur in non-IEGs (Supplementary Fig. 7c). Therefore, neuron and non-neuron cells undergo cell type dependent 5mC downregulation in IEGs, which correlate with their sustained upregulation in offspring cerebral cortex upon ZIKV-induced MIA.

Combined analysis for genome-wide 5mC profiling and bulk RNA-seq data identified 129 downregulated genes with hypermethylated features and 70 upregulated genes with hypomethylated features (Supplementary Fig. 8a). GO term analysis showed that downregulated genes are related to brain development including neural cell fate and neuronal migration while upregulated genes are linked with signaling and immune regulation (Supplementary Fig. 8b, c). In addition, DMR and DhMR annotation to transposon elements revealed that SINE element is enriched in 5hmC accumulated regions while is depleted in 5mC accumulated regions in ZIKV-affected offspring mice (Supplementary Fig. 9). Together, these results suggest that altered distribution of cell type-specific 5mC DNA methylation in IEGs correlates with their sustained mRNA upregulation in the PFC of mice exposed to MIA by ZIKV.

## Discussion

Using our established mouse model of CZS[22], we identified anxiety and depression-like emotional behavioral abnormalities in offspring mice exposed to MIA induced by ZIKV infection during the pregnancy. Through integrating bulk-seq and snRNA-seq with genome-wide 5hmC- and 5mC-profiling coupled with behavioral tests, this study defined an epigenetic mechanism by which brain methylome remodeling sustains the expression of neuronal activity genes, which could contribute to behavioral abnormalities in mice exposed to ZIKV infection during pregnancy.

We reported that anxiety and depression-like behaviors occur in mouse offspring exposed to maternal ZIKV infection. Previous studies showed that congenital ZIKV infection can result in offspring paralysis, visual and motor deficits, or spatial learning deficits[40,41]. These studies introduced the ZIKV directly into the fetal mouse brains, whereas our studies infect pregnant dams without maternal-fetal transplacental transmission of ZIKV. The abnormal behaviors in our ZIKV offspring mice are thus likely due to MIA, which is consistent with poly-functional immune activation detected in amniotic fluid from ZIKV-infected patients[42]. Emerging evidence suggests that children exposed to ZIKV during pregnancy are at increased risk of neurocognitive dysfunctions[12,13], but the scope and severity of behavioral abnormalities remain ill-defined. Our recent studies[22] showed autistic-like

behaviors including repetitive self-grooming and impaired social memory in this mouse model of MIA by ZIKV. Using the same maternal ZIKV infection paradigm, our current work identified emotional behavioral deficits. Together, these studies add significant clinical value to this ZIKV MIA animal model as we gain a comprehensive understanding of adverse cognitive and emotional deficits associated with maternal inflammation due to ZIKV infection during pregnancy.

Although a few studies on 5mC revealed differential DNA methylation associated with transcriptional changes in poly(I:C) MIA mice[21,43], 5hmC has not been examined. Our study provides the following insights into epigenetic mechanisms in maternal inflammation. First, our 5hmC profiling revealed dynamic 5hmC remodeling in ZIKV offspring brains, reflected by an overall loss of 5hmC on intragenic regions and an increase of 5hmC for IEGs. Expression of IEGs is quick and mainly transient, demanding specific gene properties such as relatively shorter length and significantly less exons. IEG genes have a high prevalence of TATA boxes and CpG islands, suggesting their specific epigenetic regulation[44]. It has been reported that IEG expression is under multiple level of epigenetic regulation, including DNA methylation[45], hydroxymethylation[46]. Overexpression and knockdown of TET protein was found to regulate specific IEG via 5hmC alteration in brain and leads to anxiety-like behavior[47]. In addition, it has been shown that the mutation of MeCP2, a well-established 5mC and 5hmC binding protein, can specifically alter transcriptional level of IEGs[48]. Global Impairment of IEG expression was also reported in Rett Syndrome patients with mutation of MeCP2[49]. Therefore, IEGs might undergo specific 5hmC regulation and show differential 5hmC patterns compared with other genes. Second, ZIKV offspring brains gain 5hmC in intergenic transposable elements (TEs). TEs are highly repetitive DNA sequences that constitute 50% of the human genome and contain ~52% of all CpG dinucleotides. Our results support the emerging view that environmental exposure during early life promotes dysregulation of TEs and increases susceptibility to neuropsychiatric disorders later in life[50]. Third, we found that 5hmC changes are associated with transcriptional changes of the corresponding genes, suggesting of a functional impact of 5hmC on gene expression. Consistent with the report that 5hmC levels positively correlate with gene expression[36,37], we identified down-regulated genes with significant loss-of-5hmC and up-regulated genes with significant gain-of-5hmC in PFC of ZIKV offspring mice. Fourth, IEG upregulation integrates and mediates neuronal activities with adaptive behaviors[28,29]. In normal circumstances, IEGs are quickly activated by neural activity and then are rapidly inactivated[28,29]. Here we showed that increased 5hmC and decreased 5mC in IEGs result in their sustained upregulation in brains of offspring mice exposed to MIA by ZIKV.

It remains to be fully determined how MIA-induced immune activation leads to lifelong neuropathology and altered behaviors in offspring. Epigenetic remodeling likely mediates the early transient changes in environments towards the stable alterations in brain functions and behaviors later in life. How the epigenetic remodeling is triggered and maintained in MIA is unclear. MIA can induce strong release of proinflammatory cytokines, such as IL-6, IL-1β, TNF-a and IL-17a[1,24]. Altered cytokine levels can lead to epigenetic changes including DNA methylation, which might occur in MIA offspring. For example, IL-6 increases the nuclear translocation of DNA cytosine-5-5methyltransferase 1 (DNMT1), the component involved in the epigenetic programming[51]; prenatal MIA causes epigenetic differences in adolescent mouse brain, including *Mecp2* promoter hypomethylation[21]. We found that ZIKV maternal infection, like the synthetic analogue of double-stranded RNA poly(I:C), induced the upregulation of a panel of immune cytokines. Future studies should provide direct evidence supporting that proinflammatory cytokines cause the epigenetic reprogramming in fetal brains in the ZIKV-induced MIA animal models. Another limitation of this study is that all the omics analyses focus on male mice. Future works are needed to

extend these analyses to females to determine the generalizability and specificity of epigenetic mechanisms of anxiety and depression behaviors.

In summary, we modeled emotional behavioral deficits in offspring mice exposed to maternal inflammation due to ZIKV infection during pregnancy. Our mechanistic study suggested an epigenetic mechanism by which brain methylome remodeling selectively alters the 5hmC and 5mC of neural activity genes, linking to emotional behaviors in ZIKV offspring mice. These results provide insights into the long-standing question of how an early transient environmental factor (maternal inflammation) is translated into genetic and circuit programs to drive long-term behaviors later in life.

# Methods

## Animals

All experimental procedures used in this study were approved by the Institutional Animal Care and Use Committee at the University of Southern California (protocol: 20719). C57BL/6 wild-type mice (Stock #: B6-F) were ordered from Taconic Biosciences. Mice were kept under specific pathogen-free (SPF) condition in a regular 12-hour light cycle (7:00 a.m. to 7:00 p.m.) at 20-24°C and controlled humidity (40-60%, usually around 50%) and provided with food and water *ad libitum*.

## ZIKA virus

ZIKV MEX1-44 was isolated in Chiapas, Mexico in January 2016 from an infected *Aedes aegypti* mosquito and was passaged by the World Reference Center for Emerging Viruses and Arboviruses (WRCEVA) in Vero cells (African green monkey kidney epithelial cells). We obtained this virus with permission through the University of Texas Medical Branch at Galveston (UTMB). As described in previous studies[15], we titrated the virus using Vero cells obtained from the American Type Culture Collection (ATCC). ZIKV stocks were generated by infecting Vero cells at a multiplicity of infection (MOI) of 0.01 and harvesting supernatants at 96 hr and 120 hr postinfection. The viral titers were determined as 50% tissue culture infectious dose ($TCID_{50}$). Briefly, supernatant fluids were serially diluted 10-fold in DMEM. A 100 μl aliquot of each diluted sample was added to 96-well plates containing a monolayer of Vero cells (three wells per dilution). After cells were cultured for 96–120 hr at 37 °C, infectious virus titers were determined by the Reed-Muench method.

## Maternal ZIKV infection

Timed-pregnant female mice (8-12 weeks old) were obtained by mating with males and the presence of seminal plugs was considered embryonic day (E) 0.5. At E12.5, pregnant female mice received an intravenous injection of $1.7 \times 10^5$ (low dosage, $ZIKV^{low}$) or $1.7 \times 10^6$ (high dosage, $ZIKV^{high}$) $TCID_{50}$ units of ZIKV suspended in 200 μl DPBS. A vehicle control group received 200 μl DPBS alone. Dams were returned to their home cages and monitored for parturition.

## Serum collection and ELISA

Pregnant female mice were anesthetized with isoflurane (induction 2.5%, maintenance 1.5%). Blood samples were collected from orbital sinus using 200 μl pipette tips at 3 hours or 48 hours after ZIKV or PBS intravenous injection. We allow the blood to clot by leaving it undisturbed at room temperature for 60 min and remove the clot by centrifuging at 2,500 g for 15 min at 4 °C. The supernatant serum was used for IL-6 (Biolegend, Cat# 431304), TNF-a (Biolegend, Cat# 430904) or IL-17a (Biolegend, Cat# 432504) ELISA measurement according to the manufacturer's protocol. Data were acquired and analyzed using SoftMAX Pro 7.1.

## Behavioral tests

All behavioral experiments were performed at 8–12 weeks of age. In all behavioral experiments, male and female offspring mice were acclimated to the behavioral testing room at least 60 minutes before the first trial began. Experimenters were blinded to animal experimental groups during behavioral tests and data analyses. The behavioral tests below were performed:

**Open field.** The subject mouse was placed in the empty arena (40 cm × 40 cm) and allowed to freely explore for 15 min. The total traveled distance and time spent in center zone were recorded and automatically measured by using Smart v3.0 (Panlab Havard Apparatus). The arena was cleaned with 75% ethanol between tests for each mouse.

**Elevated plus maze.** The subject mouse was placed in the center platform of the elevated plus maze apparatus (open arms: 25 × 5 × 0.5 cm; closed arms: 25 × 5 × 16 cm) facing the open arm and allowed to freely explore for 10 min. The number of entries and time spent in center zone, open arms, and closed arms were recorded and measured by using SMART v3.0 (Panlab Havard Apparatus).

**Rotarod.** The rotarod test consisted of training and test phases. Mice were first trained to be placed on the rotating rod at a constant speed of 4 rpm until they were able to stay on the rotating rod for 60 seconds. The test phase was performed 24 hours after the training phase. The rotarod apparatus was set to accelerate from 4 to 40 rpm in 300 seconds, and mice were placed on the rod initially rotating at 4 rpm. The latency to falling off the rod was determined. Each mouse was tested three times a day with a 15 min inter-trial interval for four consecutive days.

**Forced swimming test.** A transparent cylindrical glass container (Height × diameter: 248 mm × 184 mm) was filled with ~3500 ml distilled water to a depth such that the mouse could neither escape from the container nor touch the bottom of the container with its hind legs. The subject mouse was gently placed in the water for six minutes. The duration of immobility was recorded during the last four minutes of the test.

**Burrowing behavior.** The subject mice were habituated for one hour on the day before the burrowing assay and repeat once on the day of testing for 30 min. For acclimatization, an empty acrylic tube was placed into the home cage. The subject mice were exposed to the tube till they voluntarily enter the tube. On the day of testing, a separate cage was prepared for each mouse to be tested individually. An acrylic tube was filled with 60 g of the same corncob bedding used in the home cage and placed in one corner, parallel to the long walls of the cage. The subject mice were then transferred to these cages for 15 min, at which point the mouse was returned to the home cage and the bedding remaining in the tube was weighed. The burrowing activity was calculated by subtracting the weight of bedding present at the end of the experiment from the starting weight (60 g).

## Immunofluorescence

Animals were deeply anesthetized with 2.5% isoflurane and transcardially perfused with phosphate-buffered saline (PBS) followed by 4% paraformaldehyde. Brains were dissected out, postfixed with 4% PFA overnight at 4 °C, and dehydrated in 30% sucrose/PBS solution for 2 days. Then the samples were embedded in Tissue-Tek OCT compound (Sakura, Cat# 4583). Coronal sections were sliced at 20 μm using a cryostat (Cat# CM1950, Leica). Sections were washed in PBS three times (5 min each time) and then incubated with blocking solution (2% normal goat serum, 1% BSA, 0.3% Triton X-100 in PBS) for 2 h at room temperature. Sections were then incubated with rabbit anti-c-Fos (Cell Signaling Technology, Cat# 2250, Clone: 9F6, 1:400), rabbit anti-Egr1 (Cell Signaling Technology, Cat# 4153, Clone: 15F7, 1:200) and rabbit anti-Npas4 (Novus Biologicals, NBP2-47252, 1:200) in blocking solution overnight at 4 °C. After washing in PBS three times (5 min each

time), sections were incubated with species-specific fluorescently conjugated secondary antibodies (1:200, Invitrogen) and DAPI (Invitrogen, Cat# D21490, 1:1000) in blocking solution for 2 h at room temperature. Sections were mounted with mounting medium (Vector Laboratories, Cat# H-1000) and coverslipped. Images were captured by a Leica DMI3000 B fluorescence microscope and a Leica DMI6000 CS confocal microscope using Leica LAS X software. All images were processed and analyzed with Fiji/ImageJ software (https://fiji.sc, version 1.53 s, National Institutes of Health).

## Nuclear isolation and single-nucleus RNA-sequencing (snRNA-seq)

Male offspring mice were euthanized by $CO_2$ inhalation and then decapitated. Coronal brain slices (AP + 1.3 mm - +2.2 mm, 300 μm thickness) were cut using a vibratome (Ted Pella, Cat# 10111 N) in ice-cold dissection buffer (60 mM NaCl, 3 mM KCl, 1.25 mM $NaH_2PO_4$, 25 mM $NaHCO_3$, 115 mM sucrose, 10 mM glucose, 7 mM $MgCl_2$, 0.5 mM $CaCl_2$, 30 μM actinomycin D (Sigma-Aldrich, Cat# A1410); saturated with 5% $CO_2$ balanced $O_2$; pH = 7.4). Medial prefrontal cortices (prelimbic and infralimbic cortices) were micro-dissected under a dissection microscope and then incubated in ice-cold artificial cerebrospinal fluid (aCSF) (119 mM NaCl, 26.2 mM $NaHCO_3$, 11 mM glucose, 2.5 mM KCl, 2 mM $CaCl_2$, 2 mM $MgCl_2$, 1.2 mM $NaH_2PO_4$, 2 mM Sodium Pyruvate, 0.5 mM Vitamin C, 30 μM actinomycin D; saturated with 5% $CO_2$ balanced $O_2$; pH = 7.4) until all the samples were collected. Nuclear isolation was performed as previously described with minor modifications. Briefly, samples from four mice of each group were combined and Dounce-homogenized with four strokes of a loose pestle and four strokes of a tight pestle in ice-cold detergent lysis buffer (0.1% Triton-X, 0.32 M sucrose, 10 mM HEPES, 5 mM $CaCl_2$, 3 mM MgAc, 0.1 mM EDTA, 1 mM DTT and 30 μM actinomycin D in nuclease-free water, pH 8.0). The lysate was centrifuged at 3200x $g$ for 10 min at 4°C and the pellet was resuspended with 3 ml low sucrose buffer (0.32 M sucrose, 10 mM HEPES, 5 mM $CaCl_2$, 3 mM MgAc, 0.1 mM EDTA, 1 mM DTT and 30 μM actinomycin D in nuclease-free water, pH 8.0). The nuclei were isolated and purified by centrifugation in a sucrose density gradient at 3200x $g$ for 20 min at 4 °C, and then resuspended with resuspension solution (0.4 mg/ml BSA, 0.2 U/μl RNAse inhibitor in DPBS). Approximately 20,000 nuclei were loaded into the 10 × Chromium system with targeted recovery of 10,000 nuclei to be barcoded for snRNA-seq using a Single Cell 3' Library Kit v2 (10 × Genomics, Cat# PN-120267). Sequencing was performed on the Illumina Novaseq 6000 System. Raw read counts were analyzed using the Seurat R package.

## snRNA-seq analysis

Demultiplexing and alignment of sequencing reads to the mouse transcriptome were performed using Cell Ranger software (version 3.1.0, 10 × Genomics). We used the option -forcecells 9000" in "cell-ranger count" to extract a reasonable number of cell barcodes from samples, as we found that the automatic estimate of Cell Ranger was inaccurate. Both nuclei from ZIKV and control groups were used for clustering by Seurat (version 4.0). The top 2000 genes were identified by variable feature selection based on a variance stabilizing transformation). Then 50 principal components (PCs) were utilized to calculate a k-nearest neighbors (KNN) graph based on the Euclidean distance in PCA space and the first 30 PCs were accordingly selected for the subsequent analysis according to the Jackstraw function. Resolution in the FindClusters function was set to 1.5. Clusters were then visualized using a Uniform Manifold Approximation and Projection (UMAP) plot. To annotate the cell types by gene markers, MAST was used to perform differential gene expression analysis by comparing nuclei in each cluster to the rest of the nuclear profiles. Genes with FDR < 0.05 and log fold change ≥ 1 were selected as cell-type markers.

## Differentially expressed gene (DEG) analysis for snRNA-seq

To identify genes differentially expressed in ZIKV vs. control in each cell type, we used FindMarkers in Seurat (version 4.0) coupled with the MAST method. To estimate the relative contribution of DEGs by cell type in the ZIKV group, we downsampled the cell numbers to 50 from each cluster 10 times before performing DEG analysis. Genes with log2(fold change of expression) of at least 0.15 and FDR < 0.01 were selected as DEGs.

## Neuron and non-neuron cell isolation for genome-wide 5hmC and 5mC profiling

The neuron/non-neuron cells were isolated as previously described[38] with minor modifications. Briefly, the anesthetized male offspring mice at P21 were perfused with ice-cold 30 ml DPBS and the brains were collected. The bilateral prefrontal cortices were rapidly dissected in the ice-old HABG buffer and sliced using McIlwain tissue chopper with the thickness of 500 μm. The slices were digested in 0.2% papain in HA buffer for 30 min at 30 °C and then triturated using fire-polished Pasteur pipette. Cell suspension was applied to Optiprep density gradient and was centrifuged at 1900 rpm for 15 min at 22 °C. Neurons were collected from Fraction 3 and non-neuron cells were collected from Fraction 1 and 4. Fraction 2 was discarded.

To test the cell purity, the isolated cells were fixed with 50% ethanol for 15 min on ice. For immunofluorescent staining, two drops of neuronal or non-neuron cell suspension were applied onto glass slides and air dried. The immunostaining procedures of mouse anti-NeuN antibody (EMD Millipore, Cat# MAB377, Clone: A60, 1:400) were performed as described above. For flow cytometry, fixed cells were washed with FACS buffer (1% BSA in PBS) and blocked with Fc Block buffer (5 μg/ml, antimouse CD16/32 antibody (Biolegend, Cat# 101319, Clone: 93) in FACS uffer) for 15 min on ice. Cells were then incubated with FITC anti-NeuN antibody (Abcam, Cat# ab223994, Clone: EPR12763, 1:150) or FITC-rabbit IgG isotype control (Abcam, Cat# ab223339, Clone: EPR25A, 1:150) in FACS buffer for 30 min on ice. After two final washes in FACS buffer, the cells were fixed in 1% PFA and flow cytometry was performed on a BD LSRII. Data were collected and analyzed using BD FACDiva v9.0.

## Genome-wide 5hmC profiling

We performed 5hmC-specific chemical labeling, affinity purification and sequencing using a previously described procedure with an improved selective chemical labeling method[30]. 5hmC labeling reaction was performed in a 30 μl solution containing 1× NEB Buffer 4 (NEB, Cat# B7004S), 5 μg sonicated genomic DNA (100-500 bp), 100 μM UDP-6-N3-Glu (Jena Biosciences, Cat# CLK-076), and 62.5 U T4 phage β-Glucosyltransferase (βGT, NEB, Cat# M0357L). The reaction was incubated for 2 hours in a 37 °C water bath. DNA substrates were purified via AMPure-XP beads (BECKMAN COULTER, Cat# A63881) and reconstituted in nuclease-free water (Ambion, Cat# AM9932). Click chemistry was performed with the addition of 2 μl 2.5 mM disulfide-biotin linker (Click Chemistry, Cat# A112-10) into the DNA solution and incubated for 2 hours in a 37 °C water bath. Samples were then purified by AMPure-XP beads and DNA targets were eluted from beads by adding nuclease-free water. 5hmC-containing DNA pull-down was performed with Dynabeads MyOne Streptavidin C1 (Invitrogen, Cat# 65001) for 15 minutes at room temperature, and then 50 μl 100 mM DTT was added to release 5hmC fragments from the biotin-streptavidin bead complex by gently rotating for 2 hours at room temperature. 5hmC fragments were purified by AMPure-XP beads, DNA was eluted with nuclease-free water, and qubit quantification was performed for library preparation.

## Library preparation and sequencing

Libraries were generated following the NEBNext Ultra II DNA Library Prep Kit for Illumina (NEB, Cat# E7645) with an improved 5hmC

capture DNA library preparation method. Briefly, 5 ng of input genomic DNA or 5hmC capture DNA were used to initiate the protocol. DNA fragments were selected by AMPure-XP beads (Beckman Coulter, Cat# A63881) after NEBNext end prep and adapter ligation steps. PCR-amplified DNA libraries were quantified on an Agilent 2100 Bioanalyzer. All sequencing libraries were run on Illumina Hi-seq system by Admera Health, LLC.

## qPCR validation of 5hmC-enriched regions
5hmC captured DNA was used as templates in triplicate 20 μl qPCR reactions, mixed with 1x PerfeCTa SYBR Green FastMix, low ROX (QuantaBio, Cat# 95074), 0.25 μM forward and reverse primers, and water. Reactions were run on QuantStudio 3 System using Fast mode at 95 °C for 10 minutes, 95 °C for 15 seconds, and 60 °C for 1 minute, for 40 cycles. Fold enrichment was calculated as $2^{-dCt}$, where dCt = Ct (5-hmC enriched in ZIKV) − Ct (5-hmC enriched in Control). All primers are listed in Supplementary Table 1. Primers were designed using Primer3 online tool (https://bioinfo.ut.ee/primer3-0.4.0/) and synthesized by Integrated DNA Technologies (IDT, California, USA).

## RNA isolation and qPCR
PFC samples dissected from male offspring mice were homogenized in TRIzol (Invitrogen, Cat# 15596026) and processed according to the manufacturer's instructions. RNA was then reverse transcribed using a SuperScript III First-Strand Synthesis System (Thermo Fisher, Cat# 18080051). cDNA was quantified by qPCR using PerfeCTa SYBR Green FastMix, low ROX (QuantaBio, Cat# 95074). Each reaction was performed in triplicate and analyzed following the standard ddCt method using GAPDH as normalization control. All primers are listed in Supplementary Table 1. Primers were designed using Primer3 online tool (https://bioinfo.ut.ee/primer3-0.4.0/).

## RNA-seq
Three biological replicates (both control and ZIKV-infected mouse PFC RNA) were subjected to RNA-seq. One μg RNA sample quality was assessed by Bioanalyzer 2100 Eukaryote Total RNA Pico (Agilent Technologies, Cat# 5067-1513) and quantified by Qubit RNA HS assay (ThermoFisher, Cat# Q32852). Libraries were constructed with TruSeq Stranded mRNA library kits (Illumina, Cat# RS-122-2001) based on the manufacturer's recommendations. Library concentration was measured by qPCR and library quality evaluated by Tapestation High Sensitivity D1000 screentapes (Agilent Technologies). Equimolar pooling of libraries was performed based on qPCR values. Libraries were sequenced on a Illumina HiSeq system with a read length configuration of 150 PE targeting 40 M total reads per sample (20 M each direction) by Admera Health, LLC.

## Genome-wide 5mC profiling
We performed enzymatic methylation sequencing (EM-seq) followed by data analysis. EM-seq library construction was performed using the NEBNext Enzymatic Methyl-seq Kit (NEB, Cat# E7120L) according to manufacturer's instructions. The libraries were sequenced on Illumina Novaseq 6000 system. For EM-seq data processing, reads were filtered and trimmed with Trim Galore (version 0.4.1) followed by mapping to the mouse genome (mm9) using Bismark (version 0.17.0). Mapped reads were further deduplicated and filtered for non-conversion. Estimation of methylation levels was determined in the CpG context with Bismark. The differently methylated regions (DMRs) were identified by RADMeth (version 1.0) with an adjusted p < 0.05. DMR sites less than 100 bp away were then merged into DMR candidates. We ultimately selected candidates that contained more than 5 differently methylated CGs (among which the mean-meth-diff was > 0.2) as differently methylated regions. For CHROHMM (version 1.24) analysis, DMRs were first mapped to mm9 by liftover then mapped to existing mouse brain ChromHMM annotations using the BEDtools (version

2.28.0) intersect function. To test for enrichment of an annotation, a Fisher's exact test was performed for regions of DMRs against background.

## Methylated DNA Immunoprecipitation (MeDIP) and library construction
For each sample, 10 μg of genomic DNA was sonicated to 350 base pairs (bp) fragments using a Covaris-focused ultrasonicator and subjected to end repair, adaptor ligation, and USER digestion using the NEBNext Ultra II DNA Library Prep kit for Illumina (NEB, Cat# E7645S) according to the manufacturer's protocol. Following USER digestion and purification, DNA was denatured for 10 minutes at 95 °C and immunoprecipitated overnight at 4 °C with 4 μg of 5mC antibody (Active Motif, Cat# 39649) in MeDIP buffer (500 mM Tris-HCl pH 7.4, 750 mM NaCl and 0.25% Triton-X). The mixture was then incubated with Dynabeads Protein G (Invitrogen, Cat# 1004D) for 2 hours at 4 °C and washed with ice-cold high salt (1 M NaCl) MeIP buffer. The co-precipitated DNA fragments were finally eluted in 200 μl digest buffer (1 × TE Buffer pH 7.4, 0.25% SDS, 0.25% Proteinase K (2.5 mg/ml)) and incubated at 55 °C with shaking at 1000 rpm for 2 hours. The methylated DNA was recovered by phenol:chloroform:isoamyl alcohol (25:24:1) extraction followed by precipitation in 500 μl of 100% ethanol supplemented with 3 μl glycogen (5 mg/ml, Invitrogen, Cat# AM9510) and 15 μl sodium acetate buffer (3 M, pH 5.2, Teknova, Cat# S0297) overnight at -80 °C. The next day, the DNA was pelleted with centrifugation at 15000 rpm, washed with 75% ethanol, and dissolved in 25 μl nuclease-free water. Libraries were amplified using the NEBNext Ultra II DNA Library Prep kit for Illumina (NEB, Cat# E7645L) following the manufacturer's protocol. All libraries were quantified on an Agilent 2100 Bioanalyzer. Paired-end sequencing was performed on Illumina Hi-seq system by Admera Health, LLC.

## Bioinformatics analyses
Triplicate FASTQ sequence files from 5hmC-seq of control and ZIKV-exposed PFC were aligned to the mm9 reference genome using Bowtie2 (version 2.3.5.1). Established computational algorithms (edgeR package in R/Bioconductor environment) using quantile-adjusted conditional maximum likelihood methods were used to compare normalized 5hmC read density from three control and three ZIKV-exposed mice in 500 bp binned mouse genome. Differential regions with FDR < 0.05 were considered significant. Ngsplot (version 2.63) software was used to calculate 5hmC concatenated reads on various genomic regions and gene body. Annotation analyses of genomic features were performed using HOMER (version 4.11.1). Four replicates of RNA-seq reads from control and ZIKV-exposed mice were aligned using TopHat (version 2.1.0) and differential analyses for genes and repetitive elements were conducted by tEtranscripts (version 2.2.3). DEGs were further correlated with 5hmC changes on those genes. Gain-of-5hmC activated genes were defined as genes with logFC > 0 and gain-of-5-hmC regions (P < 0.05, logFC > 0) while loss-of-5hmC repressed genes were defined as genes with logFC < 0 and loss-of-5-hmC regions (P-value < 0.05, logFC < 0). Gene ontology analyses were performed by Gene Ontology Consortium. P-values for overlap with neuropsychiatric-linked loci were calculated using binomial tests in the R environment (version 3.6.0).

Immediate early genes (IEGs) gene body regions were defined from TSS-1000 bp to TES + 1000 bp. IEG promoter regions were defined from TSS-3000 bp to TSS + 1000 bp. The gene body regions and promoter regions were split into 500 bp continuous regions, and each 500 bp region was further split into 50 bp bins. MeDIP-seq and 5hmC-seq reads were counted in each 50 bp bin for the ZIKA and Mock groups to make a 2×10 matrix for each 500 bp region. Unpaired student's t-tests were used to determine the significant change in 5mC levels in promoter regions or 5hmC levels in gene body regions between the ZIKA and Mock groups. The combined 5hmC or 5mC level

for each bin in the ZIKA and Mock group was calculated using the following formula, assuming that 72% of cells are non-neuronal and 28% are neuronal cells in the cortex according to published studies[52]: Combined 5hmC/5mC level = 5hmC/5mC level in NeuN$^+$ × 28% + 5hmC/5mC level in NeuN$^-$ × 72%. All cell types identified by snRNA-seq were assigned to neuronal cell types including Ex1, Ex2, Ex3, Ex4, Ex5, In1, In2, In3, In4, and non-neuronal cell types including Astrocytes, Endoth, Macrophages, Microglia, OPC, and Olig. IEG expression in neuronal and non-neuronal cells was calculated by taking the average expression in all cell types for each group. Log$_2$FC of gene expression in neurons and non-neurons were then correlated with log$_2$FC of 5mC and 5hmC levels determined by MeDIP-seq and 5hmC-seq reads count as described. *Tet1* and *Actb* were non-IEG genes that served as negative controls.

### Statistic and reproducibility

Statistical analyses were performed using GraphPad Prism (version 9.0.0) software. Data are presented as mean values ± SEM. To compare more than two experimental groups, one-way ANOVA with Tukey post hoc test was performed and for comparison between two groups, two-tailed or one-tailed (as indicated in figure legends) unpaired Student's test was used to calculate *P* values. Each experiment was repeated independently at least three times with similar results. Graphs showing mean ± SEM for each group represent at least three biological replicates (as indicated in figure legends). Images of immunofluorescence staining were representative of at least three biological replicates showing similar results.

### Reporting summary

Further information on research design is available in the Nature Portfolio Reporting Summary linked to this article.

## Data availability

5hmC and RNAseq data generated in this study have been deposited in GEO under accession number GSE163268. The 5mC and snRNA-seq data generated in this study have been in deposited GEO under accession number GSE164783. The MeDIP-seq and 5hmC-seq for neuron and non-neuronal cells have been deposited into GEO under accession number GSE240417. Source data are provided with this paper.

## Code availability

The scripts utilized for the subsequent analysis of bulk RNA-seq, snRNA-seq, 5hmC-seq, and 5mC-seq datasets are available upon request.

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

## Acknowledgements

We thank Chen laboratory colleagues for stimulating discussions. We are grateful for Bridget Samuels's critical reading of the manuscript. Chen laboratory is supported by funds from the Associate Dean of Research Fund from the Center for Craniofacial Molecular Biology, Herman Ostrow School of Dentistry at the University of Southern California, as well as grants R01DE030901 (J.C.), R21AG075665 (J.C.), and R21AG070681 (J.C.) from the National Institute of Health. L. M. is a post-doctoral fellowship recipient of California Institute for Regenerative Medicine (*CIRM*) training grant.

## Author contributions

L.M., F.W., Y-P.L., J.W., Q.C., Y-N.D. conceived and performed all experiments. Z.Z., G-P.F. and B.Y. provided comments. J-F.C. wrote the manuscript with the assistance from L.M., J.S., F.W., Y-P.L, J.W., B.Y. and. J-F.C. supervised the research.

## Competing interests

The authors declare no competing interests.
