## [Peer Review File · Nature Communications]

Brain methylome remodeling selectively regulates neuronal activity genes linking to emotional behaviors in mice exposed to maternal immune activationREVIEWER COMMENTS

Reviewer #1 (Remarks to the Author):

In the study by Ma et al. a mouse model of congenital Zika virus infection is used to investigate consequences for alterations of emotional behavior in adult life. To examine potential concomitant molecular alterations the authors perform snRNAseq and methylome analysis of PFC tissues and find sustained modifications of neural IEG expression.

While the results are interesting, it remains unclear how, as stated in the discussion section" they provide "novel mechanistic insight into neuropsychiatric disorders of MIA origin". Currently, the behavioral and molecular data provided are correlative at best, but causality is not shown. Several specifics about the model itself warrant clarification before any conclusions can be drawn:

- 1.) Considering the differences between the high and low dosage model it is imperative to show the corresponding maternal immune response; especially considering the variability between different batches of viruses. Panels of proinflammatory cytokines known to be relevant in other MIA models should be analyzed in the maternal plasma.
- 2.) The sickness behavior of the dams should be recorded and this information included.
- 3.) How are rates of abortions, litter sizes, sex ratios within litters, birth weight, postnatal weight gain in the Zika group?
- 4.) Offspring of how many litters were screened for behavior?
- 5.) Offspring behavior needs to be analyzed for males and females separately and accordingly dealt with statistically
- 6.) For sequencing n=4/ group is indicated. Males or females and why?
- 7.) In the OF Zika offspring travel shorter distances, hence center time should be reported as % of time or % of distance

Reviewer #2 (Remarks to the Author):

In this manuscript, Ma et al. reported ZIKV infection-induced maternal immune activation (MIA) during mouse pregnancy influences offspring, resulting in anxiety and depression-like behavior. They performed genome-wide 5hmC and 5mC profiling using prefrontal cortex (PFC) of ZIKV offspring mice, and showed an overall loss of 5hmC DNA demethylation and an increase of 5mC DNA methylation in intragenic regions. These epigenetic changes correlate with transcriptional changes in key functional genes associated with neuropsychiatric disorders. The most important conclusion here is that a collection of IEGs are upregulated with elevated expression in excitatory neurons and associated with increased 5hmC and decreased 5mC levels in ZIKV offspring mice, contributing to the behavior defects of ZIKV offspring. Overall, this is a carefully designed study with appropriate controls and statistical analysis. The conclusion is of significance, providing a possible molecular mechanism for how environmental factors, such as MIA, could impact brain development to drive long-term behaviors later in life. I have two major suggestions and one minor comment for authors to address.

Major suggestions:

1. Both 5mC and 5hmC profiling was done using PFC that contains multiple cell types including neurons and glia cells as revealed by snRNA-seq in the manuscript. It's intriguing to see the correlation of IEGs upregulation, 5hmC enrichment, and 5mC reduction. However, it remains unclear which cell type(s) are the major affected one by ZIKV infection. As the most important conclusion in this manuscript, it will be very informative to examine 5mC and 5hmC for these IEGs using Neu+ and Neu- population isolated from PFC. Do the changes of 5mC and 5hmC agree with snRNA-seq result?
2. The author did a thoughtful integration of 5hmC and RNA-seq dataset to show the correlation between enrichment of 5hmC and transcriptional regulation. A similar analysis should be done for 5mC and RNA-seq datasets, particularly the one with change of 5mC in CpG island context.

Minor comment:

1. MIA is used in the manuscript to describe how ZIKV infection triggers the epigenome remodeling in the developing brain. Could the author define MIA at the introduction part and expand in the discussion part on how MIA does so?

Reviewer #3 (Remarks to the Author):

In this manuscript, Ma. et al. described anxiety and depression-like behaviors in mouse offspring from females experienced Zika virus infection during pregnancy. The authors dissected the transcriptomic and epigenetic changes: they demonstrated an overall increase of 5mC and a decrease of 5hmC in the prefrontal cortex, associated with expression changes in genes related to psychiatric disorders. Interestingly, the immediate early genes showed sustained expression in excitatory neurons, with a consistent increase of 5hmC. This study is timely in addressing the potential disease symptoms and molecular mechanisms of congenital Zika syndrome. The study is written and interpreted, and the mouse model is established and relevant to human disease. The molecular findings, especially the methylome remodeling and sustained IEG expression, are insightful. A few questions need to be addressed to enhance the impact:

1. A sustained increase in IEGs is a key finding here. IEGs can be induced during sample preparation – did the authors take specific measures to control this? For example, were all the control samples processed before the experimental group or the other way around?

2. For the bulk RNA-Seq analysis, $\text{abs}(\log_2 \text{ fold change}) > 0.15$ was used as the cutoff, which is lower than the commonly used thresholds. Was there a particular reason? Similarly in Fig. 5a, were the up- and down-regulated genes tested statistically?

3. The results suggest an overall decrease in 5hmC but an increase of 5hmC for IEGs. This seems to indicate the methylome was differentially regulated for the IEGs. Please describe possible mechanisms here.

Minor:

4. Please clarify whether male or female mice or both were assayed for the behavior and omics analyses.

5. Fig. 4a, please elaborate on how the blue line was fitted – it appears dominated by just a few data points. Please highlight IEGs on the plot.

6. Fig. 4c was not cited or miscited as 4d.

Sep 21, 2023

RE: NCOMMS-23-03163A

Dear Reviewers,

We are deeply appreciative for all the insightful comments and have performed substantial new experiments and thoroughly addressed individual concerns. The new data are described in "Response Letter Figure x (RL-Fig. x)" below as well as in modified Figures in the revised manuscript. Please see below for a point-by-point response to address all questions raised by the reviewers.

REVIEWER COMMENTS

Reviewer #1:

In the study by Ma et al. a mouse model of congenital Zika virus infection is used to investigate consequences for alterations of emotional behavior in adult life. To examine potential concomitant molecular alterations the authors perform snRNAseq and methylome analysis of PFC tissues and find sustained modifications of neural IEG expression. While the results are interesting, it remains unclear how, as stated in the discussion section" they provide "novel mechanistic insight into neuropsychiatric disorders of MIA origin". Currently, the behavioral and molecular data provided are correlative at best, but causality is not shown. Several specifics about the model itself warrant clarification before any conclusions can be drawn:

We agreed that the causative relationship between molecular data and behavior has not been established. Therefore, we removed the "novel mechanistic insights" in the discussion section. We appreciate all criticisms and have performed the experiments described below to address all the reviewer's questions.

1.) Considering the differences between the high and low dosage model it is imperative to show the corresponding maternal immune response; especially considering the variability between different batches of viruses. Panels of proinflammatory cytokines known to be relevant in other MIA models should be analyzed in the maternal plasma.

We appreciate this suggestion. We included a well-established MIA mouse model elicited by injection of viral mimic polyinosinic:polycytidylic acid (poly(I:C)) as positive control (PMID: 26822608, 27078638). The maternal immune response was activated by either injection of poly(I:C) or low/high dosage of ZIKV at E12.5. Serum was collected at 3 and 48 hours after injection (RL-Fig. 1a). IL-6 and TNF-a were measured at 3 hpi, both of which are reliable indicators of acute inflammatory response after MIA and serum concentration of IL-17a peaks at 48 hours after injection. Compared with PBS injection (i.v.) group as mock control, all the poly(I:C) and low/high dosage ZIKV injection successfully induced MIA, as evidenced by significantly increased serum concentrations of IL-6 and TNF-a at 3 hpi (RL-Fig. 1b and 1c), and the IL-17a also

dramatically increased at 48 hpi (RL-Fig. 1d). High dosage of ZIKV injection could induced comparable immune response as that of poly(I:C), both of which were stronger than low dosage ZIKV elicited immune response. (RL-Fig. 1b-1d).

RL-Fig. 1 (Fig. 1a-1d in revised manuscript) ZIKV infection during pregnancy elicits MIA which is comparable to those infected with poly(I:C). **a.** The schematic showing the experimental procedure for ELISA. **b-d.** ELISA measurement of serum concentrations of maternal IL-6 (**b**), TNF-a (**c**) and IL-17a (**d**). **b** and **c**, Mock $n = 5$; poly(I:C) $n=8$; ZIKV^{low} $n=5$; ZIKV^{high} $n = 7$ mice. **d**, Mock $n = 8$; poly(I:C) $n=8$; ZIKV^{low} $n=8$; ZIKV^{high} $n = 8$ mice.

2.) The sickness behavior of the dams should be recorded and this information included.

Agreed and we have performed experiments according to your suggestion. The behavioral and physiological abnormalities of sickness in pregnant dams were assessed before ZIKV injection (0 h) and at 4, 8, 12 and 24 h post-injection. Both activities in the open field and burrowing (a conserved adaptive behavior) were significantly impeded by ZIKV-elicited immune response at 4 and 8 hpi, but rapidly increased afterward till comparable to those of mock control group at 24 hpi (RL-Fig. 2a, b). The body weight of ZIKV infected dams was slightly decreased at 12 and 24 hpi and recovered after 48 hpi (RL-Fig 2c). Therefore, pregnant female mice infected with ZIKV at E12.5 exhibit transient sickness behaviors.

RL-Fig. 2 (Supplementary Fig. 1 in revised manuscript) Pregnant female mice infected with ZIKV at E12.5 exhibit transient sickness behaviors. a. Relative distance traveled in the open field (a), amount of bedding burrowed (b) and relative body weight (c) measured before injections (0) and at 4, 8, 12 and 24 h post-injection. **a**, Mock n = 14; ZIKV n = 13 mice; **b**, Mock n = 14; ZIKV n = 12 mice; **c**, Mock n = 16; ZIKV n = 18 mice.

3.) How are rates of abortions, litter sizes, sex ratios within litters, birth weight, postnatal weight gain in the Zika group?

We appreciate the reviewer's suggestion and have performed the statistical analysis of these parameters characterizing the pregnancy outcomes of ZIKV-induced MIA. Meanwhile, we also included poly(I:C) infection as positive control group. Maternal exposure to poly(I:C), low or dose of ZIKV at E12.5 did not significantly affect the pregnancy outcomes. The average litter size, birth weight, postnatal weight gain from postnatal day 0 to 28 and sex ratio of the offspring mice showed no significant difference compared with mock control group (RL-Fig. 3a-d). As shown in RL-Fig. 3e, 100% of dams of mock (15 out of 15) and low dose of ZIKV (12 out of 12) groups delivered pups, while 18% of poly(I:C) injected dams (2 out of 11) and 20% of high dose of ZIKV infected dams (3 out of 15) lost their embryos. The above observations were similar to the previous study using mouse model of poly(I:C)-induced MIA on gestation day 12 (PMID: 30026057).

RL-Fig. 3 (Supplementary Fig. 2 in revised manuscript) Pregnancy outcomes in the studies of maternal immune activation (MIA) induced by poly(I:C) and low/high dose of ZIKV on gestation day 12.5. a. The spontaneous abortion rate of dams in each treatment group. The numbers in the bar plots represent the number of dams with spontaneous abortion out of the total number of dams recorded. Mock $n = 15$; poly(I:C) $n=11$; ZIKV^{low} $n=12$; ZIKV^{high} $n = 15$ dams. **b.** The litter size delivered by dams that successfully maintained pregnancy. Mock $n = 14$; poly(I:C) $n=8$; ZIKV^{low} $n=12$; ZIKV^{high} $n = 7$ litters. **c.** The average birth weight per litter delivered by dams in each treatment group. Mock $n = 8$; poly(I:C) $n=7$; ZIKV^{low} $n=5$; ZIKV^{high} $n = 7$ litters. **d.** The average body weight of each litter delivered by dams in each treatment group from postnatal day 0 to 28. Mock $n = 7$; poly(I:C) $n=6$; ZIKV^{low} $n=7$; ZIKV^{high} $n = 9$ litters. **e.** The sex ratio of offspring born to dams that successfully maintained pregnancy. The red dashed line indicates 50% of either sex. Mock $n = 11$; poly(I:C) $n=9$; ZIKV^{low} $n=9$; ZIKV^{high} $n = 9$ litters.

4.) Offspring of how many litters were screened for behavior?

The offspring mice from 5 litters were combined for behavioral analysis.

5.) Offspring behavior needs to be analyzed for males and females separately and accordingly dealt with statistically.

We agreed and performed the experiments accordingly. To have enough numbers of male and female offspring mice for statistical analysis, we raised a new collection of ZIKV MIA mice. We repeated the behavioral tests of the mock and ZIKV group so that data from over 10 mice in each sex for each group were collected. We found that the behavioral defects of offspring exposed to maternal ZIKV infection are not sex-biased (RL-Fig. 4).

RL-Fig. 4 (Supplementary Fig. 3 and Fig. 1m in revised manuscript) The anxiety- and depression-like behaviors of offspring exposed to maternal ZIKV infection are not sex-biased. a. Total distance traveled in the open field test. Combined-Mock $n = 36$; Combined-ZIKV $n = 42$; Male-Mock $n = 19$; Male-ZIKV $n = 23$; Female-Mock $n = 17$; Female-ZIKV $n = 19$ mice. **b.** Percentage of time spent in the center area in the open field test. Combined-Mock $n = 36$; Combined-ZIKV $n = 42$; Male-Mock $n = 19$; Male-ZIKV $n = 23$; Female-Mock $n = 17$; Female-ZIKV

n = 19 mice. c. Number of entries into open arms during elevated plus maze test. Combined-Mock n = 34; Combined-ZIKV n = 43; Male-Mock n = 17; Male-ZIKV n = 24; Female-Mock n = 17; Female-ZIKV n = 19 mice. d. Time spent in the open arms during elevated plus maze test. Combined-Mock n = 34; Combined-ZIKV n = 43; Male-Mock n = 17; Male-ZIKV n = 24; Female-Mock n = 17; Female-ZIKV n = 19 mice. e. Immobility time in the forced swimming test. Combined-Mock n = 29; Combined-ZIKV n = 29; Male-Mock n = 15; Male-ZIKV n = 16; Female-Mock n = 14; Female-ZIKV n = 13 mice. f. Rotarod performance scored as time (seconds) on the rotarod. Combined-Mock n = 23; Combined-ZIKV n = 26; Male-Mock n = 12; Male-ZIKV n = 14; Female-Mock n = 11; Female-ZIKV n = 12 mice.

6.) For sequencing n=4/ group is indicated. Males or females and why?

We thank the reviewer for this detail. We used male mice for the sequencing based on the following considerations: 1) The estrous cycle in female mice can introduce additional variables that may confound the results; 2) Many early statistical and bioinformatic studies on anxiety and depression used male mice (PMID: 27181059, 29024657, 35383160). Therefore, we like to follow the established protocols to maintain consistency and comparability with previous data.

7.) In the OF Zika offspring travel shorter distances, hence center time should be reported as % of time or % of distance.

We appreciate this suggestion and have changed it accordingly.

Reviewer #2 (Remarks to the Author):

In this manuscript, Ma et al. reported ZIKV infection-induced maternal immune activation (MIA) during mouse pregnancy influences offspring, resulting in anxiety and depression-like behavior. They performed genome-wide 5hmC and 5mC profiling using prefrontal cortex (PFC) of ZIKV offspring mice, and showed an overall loss of 5hmC DNA demethylation and an increase of 5mC DNA methylation in intragenic regions. These epigenetic changes correlate with transcriptional changes in key functional genes associated with neuropsychiatric disorders. The most important conclusion here is that a collection of IEGs are upregulated with elevated expression in excitatory neurons and associated with increased 5hmC and decreased 5mC levels in ZIKV offspring mice, contributing to the behavior defects of ZIKV offspring. Overall, this is a carefully designed study with appropriate controls and statistical analysis. The conclusion is of significance, providing a possible molecular mechanism for how environmental factors, such as MIA, could impact brain development to drive long-term behaviors later in life. I have two major suggestions and one minor comment for authors to address.

Major suggestions:

1. Both 5mC and 5hmC profiling was done using PFC that contains multiple cell types including neurons and glia cells as revealed by snRNA-seq in the manuscript. It's intriguing to see the correlation of IEGs upregulation, 5hmC enrichment, and 5mC reduction. However, it remains

Jian-Fu Chen, Ph.D.
Associate Professor
Phone: 3234422062
Email: jianfu@usc.edu

unclear which cell type(s) are the major affected one by ZIKV infection. As the most important conclusion in this manuscript, it will be very informative to examine 5mC and 5hmC for these IEGs using Neu⁺ and Neu⁻ population isolated from PFC. Do the changes of 5mC and 5hmC agree with snRNA-seq result?

We appreciate this important question and have performed experiments accordingly. We employed a published density gradient centrifugation method (PMID: 17545985) to isolate NeuN⁺ and NeuN⁻ population. The purity of the isolated cells was validated by immunofluorescent staining (RL-Fig.5a) and FACS analysis (RL-Fig. 5b) using an anti-NeuN antibody. Then Methylated DNA immunoprecipitation sequencing (MeDIP-Seq) and 5hmC capture and sequencing (5hmC-seq) were conducted for each group (ZIKA NeuN⁺, ZIKA NeuN⁻, Mock NeuN⁺, Mock NeuN⁻) to evaluate the 5mC and 5hmC changes in IEGs upon ZIKA infection.

Given that it has been well established that intragenic 5hmC plays positive roles in driving gene expression, we focused on 5hmC alterations in gene bodies upon ZIKA infection. The gene bodies of IEGs were split into continuous 500 bp bins to calculate the reads count. The normalized 5hmC-seq reads count in each bin was then compared between the ZIKA and Mock groups for NeuN⁺ and NeuN⁻ cells, respectively. We found that IEGs displayed a combined 5hmC accumulation in the ZIKA group in at least one bin of their gene body (RL-Fig.5c, d), which could account for the gene upregulation shown in Fig. 2c. Among those genes, *Egr2*, *Egr3*, *Arc*, and *Fosl2* showed 5hmC accumulation in non-neuronal cells, while *Fosb*, *Npas4*, and *Nr4a1* showed 5hmC accumulation in neuronal cells. *Dusp1*, *Erg4*, and *Junb* showed 5hmC accumulation in both neuronal and non-neuronal cells. The IEG upregulation correlates with increased 5hmC in neuron or non-neuron cells (RL-Fig.5d). Together, these results suggest that neuronal and non-neuronal cells undergo cell type dependent 5hmC changes in IEGs that correlate with their sustained upregulation in offspring cerebral cortex upon ZIKV-induced MIA.

RL-Fig. 5 (Supplementary Fig. 4 in revised manuscript) 5hmC alteration in IEGs of neuron and non-neuron cells isolated from ZIKV offspring mice. a. Representative images of NeuN⁺ cells in the isolated non-neuronal (left) and neuronal fractions (right). Scale bar, 50 μ m. **b.** Histograms of NeuN-FITC fluorescence intensity in non-neuronal (left panel) or neuronal (right

panel) isolated fractions. **c.** Gene body 5hmC changes upon ZIKV infection in IEGs of neuronal (upper panel, green) and non-neuronal cells (lower panel, blue) quantified by 5hmC-seq. (Unpaired student's *t*-test, $n=10$, * $P<0.05$, ** $P<0.01$, *** $P<0.001$, **** $P<0.0001$). **d.** Log₂Fold Change of gene body 5hmC in bulk PFC (X-axis, 5hmC-seq), isolated non-neuronal (left, blue text, 5hmC-seq) and neuronal (right, green text, 5hmC-seq) fractions (Y-axis). Red dots highlight genes with significant gene body 5hmC change ($P<0.05$) in neuronal or non-neuronal cells.

In contrast to the 5hmC accumulation detected in the gene bodies, we focused on 5mC depletion in the promoter regions of those IEGs due to 5mC's established role in repressing gene transcription. Similar to our 5hmC analysis, the promoter regions of IEGs were split into 500 bp bins, and the normalized MeDIP-seq reads count for each bin was compared between the ZIKA and Mock groups for NeuN⁺ and NeuN⁻ cells, respectively. The combined 5mC levels were also calculated and compared between ZIKA and Mock groups using the same methods as we did for 5hmC analysis.

We found that IEGs showed total 5mC depletion in at least one bin of their promoter, which could contribute to the IEG upregulation observed in the ZIKA group. Among those genes, *Egr3*, *Dusp1*, *Fosb*, and *Egr2* showed 5mC depletion only in non-neuronal cells, while *Nr4a1*, *Egr1*, *Fosl2*, *Junb*, *Arc*, and *Npas4* showed 5mC depletion in both neuronal and non-neuronal cells (*RL-Fig.6a, b*). By combining 5hmC and 5mC analysis, we have found that immediate early genes showed either 5hmC or 5mC changes, or both, upon ZIKV infection, highlighting the key roles of DNA modifications in regulating their transcription.

To correlate the changes in 5mC and 5hmC with snRNA-seq, per the reviewer's suggestion, the average gene expression level of all 9 neuronal cell types and the average gene expression level of all 6 non-neuronal cell types identified in snRNA-seq were used to represent neurons and non-neurons, respectively. Consistent with their upregulation in bulk RNA-seq, we found that IEGs were upregulated in neurons and non-neuronal cells. For each IEG, we correlated the log₂FC of its expression in neurons and non-neurons with the log₂FC of 5mC and 5hmC (*RL-Fig.6c*). In summary, 3 IEGs showed both gene upregulation and 5hmC accumulation on the gene body in non-neurons (blue group), while 5 IEGs showed both gene upregulation and 5hmC accumulation on the gene body in neurons (green group), suggesting potential cell type-specific 5hmC regulatory roles in gene expression. 8 (orange group) and 4 (red group) IEGs showed gene upregulation with 5mC depletion in their promoter regions in non-neurons and neurons, respectively. Taken together, IEG upregulation is correlated with either 5hmC accumulation in the gene body or 5mC depletion in the promoter. For comparison, two non-IEGs, *Tet1* and *Actb*, were used as negative controls and showed a lack of correlation between gene expression and 5mC/5hmC (*RL-Fig.6c*). The newly generated MeDIP-seq and 5hmC-seq data has been deposited in GEO under accession number GSE240417.

Overall, the changes of 5mC and 5hmC agree with snRNA-seq result.

RL-Fig. 6 (Supplementary Fig. 7 in revised manuscript) 5mC alteration in IEGs of neuron and non-neuron cells isolated from ZIKV offspring mice. a. Promoter 5mC changes upon ZIKV infection in IEGs of neuron (upper panel, green) and non-neuron cells (lower panel, blue)

quantified by MeDIP-seq. (Unpaired student *T* test, $n=10$, $*P<0.05$, $**P<0.01$, $***P<0.001$, $****P<0.0001$). **b.** Log_2 Fold Change of promoter 5mC upon ZIKV infection in bulk PFC (Y-axis, WGBS data), isolated non-neuron (left, blue text, MeDIP-seq) and neuron (right, green text, MeDIP-seq) fractions (X-axis). Red dots highlight genes with significant promoter 5mC change ($P<0.05$) in neuron or non-neuron cells. **c.** Heatmap shows Log_2 Fold Change of average neuronal and non-neuronal gene expression from single nuclear RNA-seq data and DNA modification (5mC and 5hmC) in IEGs from NeuN⁺ or NeuN⁻ cell populations. *Actb* and *Tet1* were not IEGs and served as negative controls.

2. The author did a thoughtful integration of 5hmC and RNA-seq dataset to show the correlation between enrichment of 5hmC and transcriptional regulation. A similar analysis should be done for 5mC and RNA-seq datasets, particularly the one with change of 5mC in CpG island context.'

We appreciate the reviewer's suggestion and performed the corresponding analysis. Homer was used to annotate 346 hypermethylated and 169 hypomethylated sites. Unlike Homer annotation for DhMRs (Fig. 4e), hypomethylated regions were mainly annotated to intron and intergenic region while hypermethylated regions were more annotated to promoter (RL-Fig. 7). Combined analysis for genome-wide 5mC profiling and bulk RNA-seq data identified 129 downregulated genes with hypermethylated features and 70 upregulated genes with hypomethylated features (RL-Fig. 8a). GO term analysis showed that downregulated genes are related to brain development including neural cell fate and neuronal migration while upregulated genes are linked with signaling and immune regulation (RL-Fig. 8b, c). In addition, DmR and DhMR annotation to transposon elements revealed that SINE element is enriched in 5hmC accumulated regions while is depleted in 5mC accumulated regions in ZIKV-affected offspring mice (RL-Fig. 9). Together, these results suggest that altered distribution of cell type-specific 5mC DNA methylation in IEGs correlates with their sustained mRNA upregulation in the PFC of mice exposed to MIA by ZIKV.

RL-Fig. 7 (Supplementary Fig. 6b and Fig 4e in revised manuscript) Genomic distribution feature for DmR (a) and DhMR (b) in ZIKV-exposed offspring mice.

RL-Fig. 8 (Supplementary Fig. 8 in revised manuscript) Gene expression changes with concomitanted change of 5mC alteration in the PFC of offspring mice exposed to maternal ZIKV infection. **a.** Gene expression change in PFC of offspring mice exposed to maternal ZIKV infection. Genes with concomitanted change of 5mC and expression are highlighted in blue and red respectively. **b-c.** Functional enrichment of genes display concomitanted down (**b**) or up (**c**) regulation of both 5mC and expression.

RL-Fig. 9 (Supplementary Fig. 9 in revised manuscript) Enrichment analysis of DmR and Dhmr identified from the PFC of offspring mice exposed to maternal ZIKV infection to TE families.

Minor comment:

1. MIA is used in the manuscript to describe how ZIKV infection triggers the epigenome remodeling in the developing brain. Could the author define MIA at the introduction part and expand in the discussion part on how MIA does so?

Yes. We defined MIA at the introduction and expand it in the discussion. Maternal immune activation (MIA) refers to as activation of a pregnant woman's immune system during pregnancy, which can be triggered by various factors, including infections, autoimmune diseases, or exposure to certain environmental factors. It remains to be fully determined how MIA-induced immune activation leads to lifelong neuropathology and altered behaviors in offspring. Epigenetic remodeling likely mediates the early transient changes in environments towards the stable alterations in brain functions and behaviors later in life. MIA can induce strong release of proinflammatory cytokines, such as IL-6, IL-1 β , TNF- α and IL-17a (PMID: 27540164, 26822608). Altered cytokine levels can lead to epigenetic changes including DNA methylation, which might occur in MIA offspring 25180573). For example, IL-6 increases the nuclear translocation of DNA cytosine-5-methyltransferase 1 (DNMT1), the component involved in the epigenetic programming (PMID: 18204201); Prenatal MIA causes epigenetic differences in adolescent mouse brain, including Mecp2 promoter hypomethylation (PMID: 25180573). We found that ZIKV-induced MIA, like the synthetic analogue of double-stranded RNA poly(I:C), induced the upregulation of a panel of immune cytokines. Future studies should provide direct evidence supporting that proinflammatory cytokines cause the epigenetic reprogramming in fetal brains in the ZIKV-induced MIA animal models.

Reviewer #3 (Remarks to the Author):

In this manuscript, Ma. et al. described anxiety and depression-like behaviors in mouse offspring from females experienced Zika virus infection during pregnancy. The authors dissected the transcriptomic and epigenetic changes: they demonstrated an overall increase of 5mC and a decrease of 5hmC in the prefrontal cortex, associated with expression changes in genes related to psychiatric disorders. Interestingly, the immediate early genes showed sustained expression in excitatory neurons, with a consistent increase of 5hmC. This study is timely in addressing the potential disease symptoms and molecular mechanisms of congenital Zika syndrome. The study is written and interpreted, and the mouse model is established and relevant to human disease. The molecular findings, especially the methylome remodeling and sustained IEG expression, are insightful. A few questions need to be addressed to enhance the impact:

1. A sustained increase in IEGs is a key finding here. IEGs can be induced during sample preparation – did the authors take specific measures to control this? For example, were all the control samples processed before the experimental group or the other way around?

We thank the reviewer for the attention to this detail. We have added transcription inhibitor actinomycin D (Sigma-Aldrich, Cat# A1410) at the concentration of 30 μ M into all the solutions used during the nuclei isolation in order to minimize the dissociation-induced IEG expression. We have added this information in our “Methods” session.

2. For the bulk RNA-Seq analysis, $\text{abs}(\log_2 \text{ fold change}) > 0.15$ was used as the cutoff, which is lower than the commonly used thresholds. Was there a particular reason? Similarly in Fig. 5a, were the up- and down-regulated genes tested statistically?

Thank you for these questions. For Fig. 2a in the manuscript, we used gene expression level >150 and $|\log_2 \text{ Fold Change}| > 0.15$ as cutoff to get basal level high genes with expression changes in ZIKA offspring using Cuffdiff, which is consistent with the cutoff we used for snRNA-seq analysis. All IEG genes that we focused on in this manuscript were validated by RT-qPCR (Fig. 2c) and show statistically significance ($P\text{-value} < 0.05$). In Fig. 5a, all genes that highlighted display significantly changes in 5hmC level ($FDR < 0.05$) while RNA-seq $\log_2 \text{ Fold Change}$ calculated by TETranscripts (Software) were used to indicate that the expression change is positively correlated with 5hmC change. To make it consistent, we updated Fig. 2a-b using TETranscripts result (RL-Fig. 10).

RL-Fig. 10 (Fig. 2a-2b in revised manuscript) Differential gene expression analysis and DE genes functional enrichment. a. Log₂-fold change and average expression level of all genes quantified by bulk RNA-seq reads in the PFC of ZIKV offspring mice. **b.** Gene ontology analysis for significantly upregulated genes in ZIKV-affected offspring mice.

3. The results suggest an overall decrease in 5hmC but an increase of 5hmC for IEGs. This seems to indicate the methylome was differentially regulated for the IEGs. Please describe possible mechanisms here.

We thank the reviewer for this insightful comment. We have discussed possible mechanisms in the discussion section. Expression of IEGs is quick and mainly transient, demanding specific gene properties. Compare with other genes, IEGs have on average shorter length and significantly fewer exons. More importantly, they have high prevalence of TATA boxes and CpG islands, suggesting their specific epigenetic regulation (PMID: 22983151). Indeed, it has been reported that IEG expression is under multiple level of epigenetic regulation, including DNA methylation (PMID: 30545945, 27553230), hydroxymethylation⁴ and histone acetylation (PMID: 27553230). For example, overexpression and knockdown of TET protein was found to regulate specific IEG via 5hmC alteration in brain and leads to anxiety-like behavior (PMID: 32103150, 28899881). In addition, it has been shown that mutation of MeCP2, a well-established 5mC and 5hmC binding protein (PMID: 24766768, 23260135), can specifically alter transcriptional level of IEG (PMID: 22395464). Global Impairment of IEG expression was also reported in Rett Syndrome patients with mutation of MeCP2 (PMID: 36674969). Taken together, IEGs could undergo specific 5hmC regulation thus show differentially 5hmC patterns compare with other genes.

Minor:

4. Please clarify whether male or female mice or both were assayed for the behavior and omics analyses.

We appreciate this comment. In our manuscript, the male and female mice were combined for the behavior test as previous studies (PMID: 33526942, 35368297). These results showed that

C57BL/6N mice did not exhibit sex differences in anxiety- and depressive- like behaviors. During the revision, we raised an independent cohort mice with or without maternal ZIKV infection and repeated the behavior tests with over 10 male or female mice for both groups and further confirmed no sex effects on behavioral phenotypes of our ZIKV-induced MIA mouse model (RL-Fig. 11),

For omics analyses, we exclusively used male mice based on the following considerations: 1) The estrous cycle in female mice can introduce additional variables that may confound the results; 2) Many early bioinformatic studies on anxiety and depression used male mice (PMID: 27181059, 29024657, 35383160). We like to follow established protocols to maintain consistency and comparability with existing data. We have added this information in our revised manuscript.

RL-Fig. 11 (Fig. S3 in revised manuscript) The anxiety- and depression-like behaviors of offspring exposed to maternal ZIKV infection are not sex-biased. **a.** Total distance traveled in the open field test. Male-Mock $n = 19$; Male-ZIKV $n = 23$; Female-Mock $n = 17$; Female-ZIKV $n = 19$ mice. **b.** Percentage of time spent in the center area in the open field test. Male-Mock $n = 19$; Male-ZIKV $n = 23$; Female-Mock $n = 17$; Female-ZIKV $n = 19$ mice. **c.** Percentage of number of entries into open arms during elevated plus maze test. Male-Mock $n = 17$; Male-ZIKV $n = 24$; Female-Mock $n = 17$; Female-ZIKV $n = 19$ mice. **d.** Percentage of time spent in the open arms during elevated plus maze test. Male-Mock $n = 17$; Male-ZIKV $n = 24$; Female-Mock $n = 17$; Female-ZIKV $n = 19$ mice. **e.** Immobility time in the forced swimming test. Male-Mock $n = 15$; Male-ZIKV $n = 16$; Female-Mock $n = 14$; Female-ZIKV $n = 13$ mice. **f.** Rotarod performance scored

Jian-Fu Chen, Ph.D.
Associate Professor
Phone: 3234422062
Email: jianfu@usc.edu

as time (seconds) on the rotarod. Male-Mock $n = 12$; Male-ZIKV $n = 14$; Female-Mock $n = 11$; Female-ZIKV $n = 12$ mice.

5. Fig. 4a, please elaborate on how the blue line was fitted – it appears dominated by just a few data points. Please highlight IEGs on the plot.

We appreciate this suggestion and revise it accordingly. The blue trend line is generated by function `geom_smooth` using linear model in `ggplot2` based on all point plotted in the figure. The detailed command used is “`geom_smooth(method = "lm", se=TRUE,color="blue", formula = y ~ x)`”. Per reviewer’s suggestion, we have replaced Fig. 4a with scatter plot highlighting all 6 loci shown in Fig. 5g and h.

6. Fig. 4c was not cited or miscited as 4d.

We apologize for the mistake and have corrected it accordingly.

We greatly appreciate all the reviewers’ comments and are confident that these revisions have significantly improved this study. Thank you very much for your help!

Your sincerely,

Jianfu Chen

Associate Professor

Center for Craniofacial Molecular Biology

Division of Biomedical Sciences, Ostrow School of Dentistry

University of Southern California (USC)

2250 Alcazar St. CSA 135

Los Angeles, Ca 90033

Phone (323) 442-2062

Email: jianfu@usc.edu

REVIEWERS' COMMENTS

Reviewer #1 (Remarks to the Author):

The authors have done a very good job in this revision and been responsive to all concerns raised, including the addition of required further experiments.

I am ready to suggest acceptance of the manuscript in the present version.

Reviewer #2 (Remarks to the Author):

The authors did a series of new experiments using NeuN+ and NeuN- population to address my questions, and performed new analysis to deepen the interpretation and discussion of the result. The conclusion of this manuscript is of significance to the broader readers. I recommend it to be accepted and published.

Reviewer #3 (Remarks to the Author):

The authors have addressed my previous comments and concerns. I recommend publication.

Point-by-point response to reviewers' comments

Manuscript: [NCOMMS-23-03163A]

Reviewer #1:

Remarks to the Author:

The authors have done a very good job in this revision and been responsive to all concerns raised, including the addition of required further experiments.

I am ready to suggest acceptance of the manuscript in the present version.

Response: We thank the reviewer for the careful and insightful review of our manuscript, which significantly improve this study.

Reviewer #2:

Remarks to the Author:

The authors did a series of new experiments using NeuN+ and NeuN- population to address my questions, and performed new analysis to deepen the interpretation and discussion of the result. The conclusion of this manuscript is of significance to the broader readers. I recommend it to be accepted and published.

Response: We thank the reviewer very much for the review and the opinion that the revisions we made to the manuscript were satisfying.

Reviewer #3:

Remarks to the Author:

The authors have addressed my previous comments and concerns. I recommend publication.

Response: We thank the reviewer for the positive evaluation. The constructive criticism helped us to improve our study and bring it to its current state.